# Hallucination at a Glance: Controlled Visual Editing and Fine-Grained Multimodal Learning

**Tianyi Bai**[1,2*]**, Yuxuan Fan**[3*]**, Qiu Jiantao**[2*†]**, Fupeng Sun**[4]**, Jiayi Song**[5]**, Junlin Han**[6]**,**
**Zichen Liu**[1]**, Conghui He**[2‡]**, Wentao Zhang**[5,2‡]**, Binhang Yuan**[1‡]

[1]The Hong Kong University of Science and Technology
[2]Shanghai Artificial Intelligence Laboratory
[3]The Hong Kong University of Science and Technology (Guangzhou)
[4]Imperial College London, [5]Peking University, [6]Oxford University
`heconghui@pjlab.org.cn, wentao.zhang@pku.edu.cn, biyuan@ust.hk`

## Abstract

Multimodal large language models (MLLMs) have achieved strong performance on vision-language tasks but still struggle with fine-grained visual differences, leading to hallucinations or missed semantic shifts. We attribute this to limitations in both training data and learning objectives. To address these issues, we propose a controlled data generation pipeline that produces minimally edited image pairs with semantically aligned captions. Using this pipeline, we construct the Micro Edit Dataset (MED), containing over 50K image-text pairs spanning 11 fine-grained edit categories, including attribute, count, position, and object presence changes. Building on MED, we introduce a supervised fine-tuning (SFT) framework with a feature-level consistency loss that promotes stable visual embeddings under small edits. We evaluate our approach on the Micro Edit Detection benchmark, which includes carefully balanced evaluation pairs designed to test sensitivity to subtle visual variations across the same edit categories. Our method improves difference detection accuracy and reduces hallucinations compared to strong baselines, including GPT-4o. Moreover, it yields consistent gains on standard vision-language tasks such as image captioning and visual question answering. These results demonstrate the effectiveness of combining targeted data and alignment objectives for enhancing fine-grained visual reasoning in MLLMs. Code and datasets are publicly released at `https://github.com/Relaxed-System-Lab/hallu_med`.

## 1 Introduction

Multimodal large language models (MLLMs) have achieved impressive results on a wide range of vision-language tasks, including visual question answering, image captioning, and referring expression comprehension [8]. Despite this progress, such MLLMs sometimes become strikingly brittle when it comes to fine-grained visual understanding—the ability to detect and reason over small but semantically meaningful changes in images [46]. As shown in Figure 1, even state-of-the-art models like GPT-4o frequently generate fluent but incorrect responses when faced with minimal edits involving object presence, count, spatial position, or attributes.

This limitation presents an obstacle for the deployment of MLLMs in real-world applications that demand high precision. Domains such as robotics [9], industrial quality control [16], medical

---

[*]Contribute equally.

[†]Project leader.

[‡]Corresponding authors.

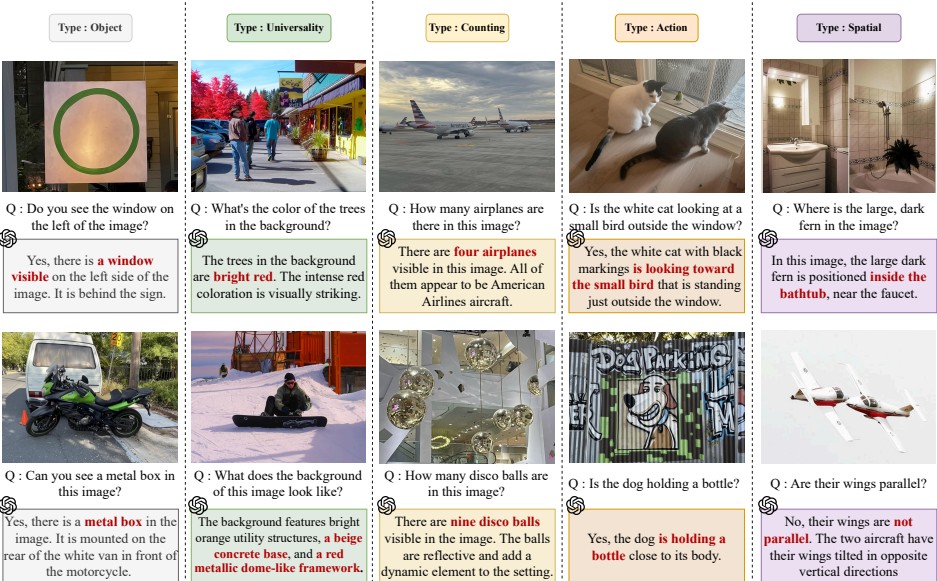

| Type : Object | Type : Universality | Type : Counting | Type : Action | Type : Spatial |
|---|---|---|---|---|
| Q : Do you see the window on the left of the image? | Q : What's the color of the trees in the background? | Q : How many airplanes are there in this image? | Q : Is the white cat looking at a small bird outside the window? | Q : Where is the large, dark fern in the image? |
| Yes, there is **a window visible** on the left side of the image. It is behind the sign. | The trees in the background are **bright red**. The intense red coloration is visually striking. | There are **four airplanes** visible in this image. All of them appear to be American Airlines aircraft. | Yes, the white cat with black markings **is looking toward the small bird** that is standing just outside the window. | In this image, the large dark fern is positioned **inside the bathtub**, near the faucet. |
| Q : Can you see a metal box in this image? | Q : What does the background of this image look like? | Q : How many disco balls are in this image? | Q : Is the dog holding a bottle? | Q : Are their wings parallel? |
| Yes, there is a **metal box** in the image. It is mounted on the rear of the white van in front of the motorcycle. | The background features bright orange utility structures, **a beige concrete base, and a red metallic dome-like framework.** | There are **nine disco balls** visible in the image. The balls are reflective and add a dynamic element to the setting. | Yes, the dog **is holding a bottle** close to its body. | No, their wings are **not parallel**. The two aircraft have their wings tilted in opposite vertical directions |

Figure 1: **GPT-4o exhibits hallucination errors on fine-grained visual question answering.** Instances are systematically selected to highlight failure cases in GPT-4o (Date accessed: May 08, 2025) visual question answering (VQA) performance across five hallucination-prone categories [15, 31]: Object, Universality, Counting, Action, and Spatial. Each row shows two examples where GPT-4o provides fluent yet inaccurate answers due to subtle misinterpretation of visual details. Hallucinated content is shown in red bold text, indicating model-generated descriptions that are not grounded in the image. All images are sourced from DOCCI and Visual Genome.

imaging [45], and assistive AI [63] all require reliable grounding in subtle visual cues. Crucially, the distinctions that current models fail to capture—such as whether a tool is present, how many objects are in view, or the direction an object is facing—are trivial for humans, suggesting a fundamental mismatch between MLLM representations and the demands of fine-grained reasoning [4, 15].

We attribute this deficiency to two intertwined factors: the lack of suitable training data and the limitations of current learning objectives. Large-scale web-crawled datasets rarely contain image pairs with tightly controlled, minimal semantic differences and aligned textual descriptions, making it difficult for models to learn how small visual changes map to linguistic shifts [65] . Moreover, existing training paradigms do not explicitly enforce feature-level stability across such small perturbations, resulting in brittle visual-textual alignments that easily drift under fine-grained edits [40].

To address these issues, we propose a framework that combines targeted data construction with a new fine-grained alignment objective. We introduce a semantically controlled image editing pipeline that generates minimally modified image pairs with precise, contrastive captions. Using this pipeline, we construct the Micro Edit Dataset (MED)—a large-scale dataset comprising over 50K image-text pairs across 11 fine-grained edit types, including attribute changes, object insertion/removal, and so on.

To leverage this data, we design a supervised fine-tuning (SFT) strategy that incorporates a feature consistency regularization term, which encourages the image encoder to produce stable embeddings across visually similar inputs. This objective aligns visual representations more closely with semantic granularity, helping reduce hallucinations and improve robustness.

Finally, we introduce the Micro Edit Detection benchmark, a new evaluation suite designed to directly assess model sensitivity to subtle visual changes. Empirical results show that our method significantly outperforms strong baselines—including GPT-4o—on both edit detection and standard VQA and captioning tasks, demonstrating improved grounding and reduced hallucination.

Our contributions are as follows:

- We introduce a semantically controlled image editing pipeline to generate high-quality, minimally different image-caption pairs at scale.

- We construct and release the Micro Edit Dataset (MED) and the Micro Edit Detection benchmark, targeting fine-grained vision-language reasoning.
- We propose a feature consistency regularization objective that improves representational stability under small semantic edits.
- We demonstrate that our approach reduces hallucinations and improves fine-grained visual understanding across multiple benchmarks and models.

## 2 Related Work

**Image editing.** Image editing involves modifying the visual appearance, structure, or elements of an existing image [23]. While GANs [18, 27, 44] pioneered realistic image manipulation, recent diffusion models [22, 48, 50] and flow-based models [35, 32] have further advanced visual generation and editing capabilities. To achieve more controllable and guided editing, various approaches have been developed, leveraging modalities such as textual instructions [66, 7, 6], masks [24, 56], layouts [13, 37], segmentation maps [39, 59], strokes [41, 58, 38], references [11, 12], and point-dragging interfaces [42, 44]. Recent state-of-the-art image editing models enable precise localized modifications while preserving the source content and maintaining consistency with the original distribution [5, 34, 19]. Leveraging these advances, we use the Gemini Flash 2.0 model [19] to generate image pairs with subtle local differences, forming the foundation of our dataset.

**Multimodal LLMs.** Multimodal large language models (MLLMs) integrate vision encoders with pretrained language models, enabling joint reasoning over text and images, as seen in [55, 33]. These typically combine visual encoders like CLIP [47] with adapter modules (MLPs, query-based transformers, or attention) to connect modalities. Discrete-token models such as Emu [51, 52, 57] and Chameleon [53] process all modalities as token sequences in a single transformer. Despite advancements, MLLMs still struggle with fine-grained visual understanding, often hallucinating details or missing subtle differences, especially in minimal-difference tasks [29, 15]. These challenges arise from datasets lacking controlled variability and misaligned training objectives. Our work seeks to address these issues by improving data quality and training methods to enhance fine-grained reasoning in MLLMs.

**Evaluating Multimodal LLMs** The evaluation of Multimodal Large Language Models (MLLMs) has progressed from initial benchmarks like VQAv2 [20], GQA [25], and TextVQA [49] to newer frameworks such as MM-Vet [61, 62], POPE [30], MMBench [36], MMStar[10] and MMVP [64], which focus on robustness, factual alignment, and hallucination. However, most benchmarks still emphasize coarse-grained tasks and overlook models' struggles with subtle, semantically important visual differences—crucial for precision-sensitive use cases [14]. Fine-grained visual understanding remains a significant challenge, with MLLMs often missing minor but meaningful changes, resulting in hallucinations or errors. To address this, we introduce the Micro Edit Detection benchmark, which evaluates MLLMs on reasoning over minimally different image pairs, thereby complementing existing benchmarks and enabling more precise assessment of visual grounding and robustness.

## 3 The Micro Edit Dataset Construction

To advance fine-grained visual understanding in MLLMs, we construct a dataset focused on subtle semantic differences, addressing gaps in existing benchmarks. Using DOCCI [43] and Visual Genome [28], we develop a pipeline with filtering, semantic edit planning, and controlled image editing. In Section 3.2, we detail the complete pipeline used to generate the 50K-image Micro Edit Dataset, including the selection criteria, edit taxonomy, and caption alignment strategy. In Section 3.3, we describe how we construct the Micro Edit Detection (MED) benchmark from this dataset, which serves as a targeted evaluation tool for assessing fine-grained reasoning capabilities in MLLMs.

### 3.1 Data Generation

To construct a dataset for evaluating fine-grained visual understanding, we sample candidate image-text pairs from DOCCI and Visual Genome, which offer diverse, object-rich images and detailed captions—ideal for producing semantically minimal image pairs.

**MLLM-Guided Data Filtering.** Due to quality and editability variations in raw image-caption pairs from DOCCI and Visual Genome, we employ a filtering pipeline using `Qwen-2.5-VL-72B`, a robust pretrained MLLM, for automatic evaluation. The model evaluates image quality, caption clarity, and editability using a structured template, considering factors such as sharpness, subject prominence,

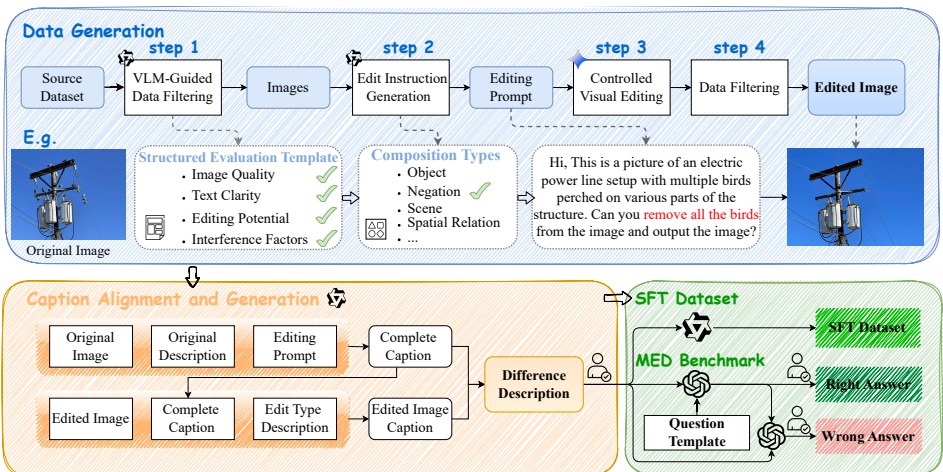

Figure 2: **Overview of the Micro Edit Dataset (MED) construction pipeline.** We begin with MLLM-guided filtering and controlled visual editing based on hallucination-prone change types. Caption alignment is performed via step-wise prompting to ensure consistency between original, edited images, and textual descriptions. The resulting data supports both supervised fine-tuning and benchmark evaluation for fine-grained visual reasoning.

caption specificity, and whether minor edits would alter caption meaning. Only samples receiving a strict "Yes" are retained, ensuring alignment between visuals and text for semantic-preserving edits.

**Controlled Visual Editing.** After filtering, minimally different image pairs are generated using `Gemini Flash 2.0` for precise edits, guided by edit instructions created by `Qwen-2.5-VL-72B`. These instructions are based on eleven compositional change categories linked to common MLLM hallucination sources [31, 15], such as object manipulation, attribute changes, spatial reconfiguration, and complex edits like counting and comparison (see Figure 3). For each image, the most suitable category—favoring non-object edits—is chosen, and a fixed-format natural language prompt is generated. These instructions ensure minimal yet meaningful changes, resulting in contrastive image pairs with aligned original and updated captions reflecting the visual edits.

**Data Filtering.** To ensure the generated image pairs differ only in subtle ways, we introduce a visual similarity filtering stage inspired by [54]. Using CLIP embeddings from the `clip-vit-base-patch32` model, we calculate cosine similarity and discard pairs below a 0.7 threshold following the MMVP [64]. This filtering retains pairs with subtle, meaningful edits, emphasizing minimal visual differences and supporting fine-grained visual understanding evaluation.

## 3.2 Caption Alignment and Generation

To ensure image pairs have semantically precise and visually grounded captions, we design a structured caption generation pipeline shown in 2. This pipeline both verifies caption correctness after editing and removes inconsistent image-text pairs. Although automatic fine-grained alignment is difficult, we leverage state-of-the-art MLLMs for captioning and text comparison. Specifically, we break the alignment check into four steps, each handled independently by `Qwen2.5-VL-72B`.

**Step 1: Caption Completion for Original Image.** Original captions in DOCCI and Visual Genome are often brief and may omit elements relevant to the edit prompt. To create a faithful reference, we prompt the model with the original image, its caption, and the edit prompt, asking it to revise the caption only if the image visibly includes the specified elements. This ensures the original caption is comprehensive yet strictly grounded in the visible content.

**Step 2: Caption Generation for Edited Image.** With the edited image in hand, we generate a new caption that accurately reflects the updated content. The model is given the complete original caption as an example, and a description of the editing type (e.g., "change in spatial relation") to focus its attention on the most relevant aspects.

**Step 3: Generating Difference Description.** For contrastive supervision, we compare captions from Step 1 and Step 2, then use `Qwen3-32B`, a strong text-only LLM, to generate a concise description of their key difference. The model identifies the most salient change and classifies it into a predefined

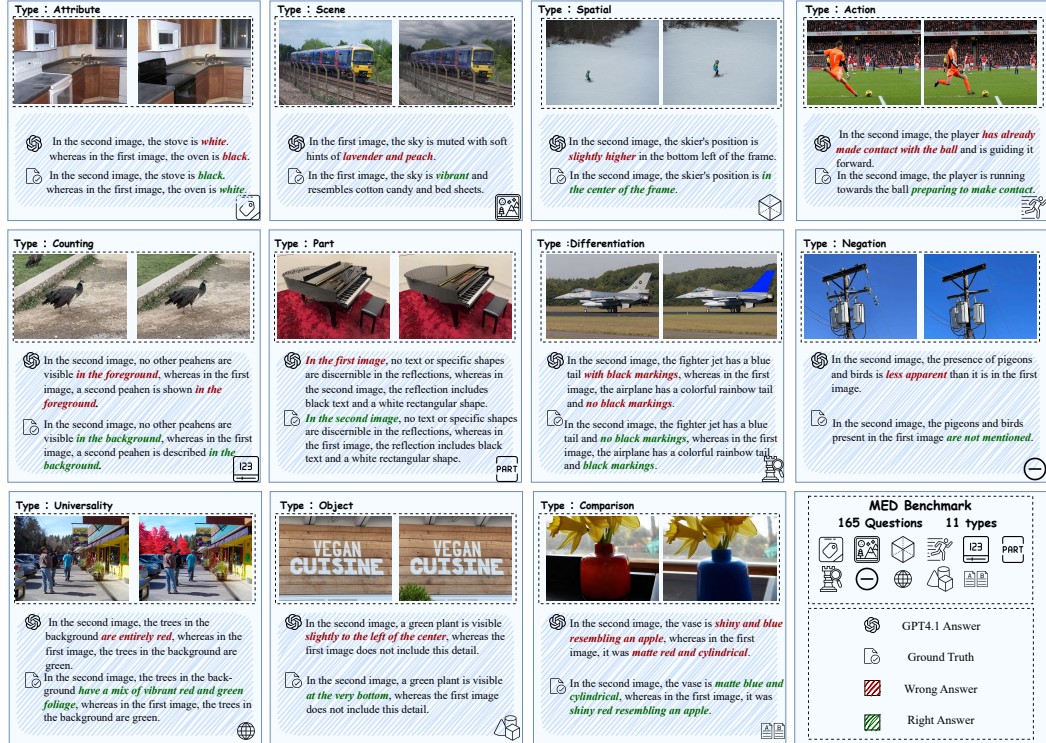

Figure 3: **Examples from the MED Benchmark covering all 11 composition types**. These categories reflect common sources of hallucination in MLLMs [15, 31] and include both basic changes (Object, Attribute, Scene, Spatial, Action, Part) and more complex transformations (Counting, Comparison, Differentiation, Negation, Universality). Each example shows an image pair, a difference question, and multiple-choice answers including the correct response and plausible distractors.

semantic edit type. This difference sentence is later used during training or evaluation to support fine-grained reasoning over minimal edits.

**Step 4: Human Verification and Iterative Refinement.** To validate data quality, we manually inspect 1,000 sampled image pairs across all eleven visual change types. Annotators evaluate whether captions and difference descriptions accurately align with the visual content. Based on issues found—like under-specified changes or inaccurate attributes—we refine prompts and reprocess affected samples using Qwen-VL-Max. This helps eliminate systematic errors and ensures high data quality. To enhance difference descriptions, `Qwen3-32B` and `Qwen2.5-VL-72B` rewrite each sample with richer vocabulary and improved clarity. Visual changes are rephrased into a QA format, forming MED, an instruction dataset with over 50K high-quality samples paired with the images.

### 3.3 The Micro Edit Detection (MED) Benchmark

Using MED, we construct a benchmark for evaluating fine-grained visual reasoning in multimodal models. We select image pairs with high semantic similarity (CLIP cosine similarity > 0.95), and for each pair, generate nine rephrased questions (see Appendix D). `GPT-4o` produces the correct answer and three subtle distractors for each question. All question-answer sets are manually verified and categorized into eleven edit types. To prevent contamination, benchmark samples are excluded from the fine-tuning dataset. The final benchmark contains 165 questions evenly distributed across all edit types, offering a robust tool for fine-grained visual reasoning evaluation.

To address potential concerns about overfitting to synthetic data, as both the training set and benchmark stem from the same controlled editing pipeline, we augment the evaluation with real-world image pairs. The MED-Real Set is created by sampling 50 minimally different image pairs from the MMVP benchmark [54], selecting those with CLIP similarity scores above 0.95 to capture subtle yet meaningful visual variations. Question-answer items are generated for each pair using the same fine-grained reasoning format as in the main benchmark. This expands the evaluation set to 215

items, combining 165 synthetic edit pairs and 50 real-world pairs, offering a more comprehensive assessment of sensitivity to controlled differences and real-world generalization.

# 4 Fine-Grained Multimodal Learning for MLLMs

In this section, we present our approach for enhancing fine-grained visual difference understanding in MLLMs. Section 4.1 formalizes the problem setting. Section 4.2 analyzes the impact of semantic misalignment and objective mismatch on downstream performance. Finally, Section 4.3 introduces a supervised fine-tuning (SFT) strategy that leverages semantically controlled data augmentation to improve the MLLM's capacity for fine-grained multimodal reasoning.

## 4.1 Problem Setup

We formalize the task of fine-grained visual difference detection using MLLMs. Let $\mathcal{X}$ denote the space of images, equipped with a perceptual premetric $d_{\mathcal{X}}$ (e.g., Learned Perceptual Image Patch Similarity (LPIPS)). Let $\mathcal{T}$ denote the space of natural language captions, equipped with a semantic premetric $d_{\mathcal{T}}$ (e.g., edit distance or BERT-score). Define the ground truth difference operator $\Delta_{\mathcal{T}}(t_1, t_2)$ as the natural language description of the difference between two captions $t_1$ and $t_2$. A MLLM $\mathsf{M}_{\theta} = \{\mathsf{I}_{\theta}, \mathsf{T}_{\theta}, \mathsf{Z}_{\theta}\}$ consists of an image encoder $\mathsf{I}_{\theta}$, a text encoder $\mathsf{T}_{\theta}$, and an text decoder $\mathsf{Z}_{\theta}$, with parameters $\theta$. We define two loss functions: the autoregressive captioning loss $l_{\text{cap}}$ and a CLIP-style contrastive loss $l_{\text{clip}}$. Given a distribution $\mathcal{P}(\mathcal{X})$ and a ground-truth captioning function $f : \mathcal{X} \to \mathcal{T}$, MLLM training aims to minimize the population risk (equation (4) in [60]):

$$R(\theta) = \mathbb{E}_{x \sim \mathcal{P}(\mathcal{X}), \, t=f(x)} \left[ \lambda_{\text{cap}} \cdot l_{\text{cap}}(\mathsf{Z}_{\theta}[\mathsf{I}_{\theta}(x)], t) + \lambda_{\text{clip}} \cdot l_{\text{clip}}(\mathsf{I}_{\theta}(x) - \mathsf{T}_{\theta}(t)) \right], \quad (1)$$

where $\lambda_{\text{cap}}$ and $\lambda_{\text{clip}} > 0$ are penalty parameters.

The empirical risk[4] over a dataset $\mathcal{D} = \{(x_i, t_i)\}_{i=1}^{N}$ is:

$$\hat{R}(\theta) = \frac{1}{|\mathcal{D}|} \sum_{(x_i, t_i) \in \mathcal{D}} \left[ \lambda_{\text{cap}} \cdot l_{\text{cap}}(\mathsf{Z}_{\theta}[\mathsf{I}_{\theta}(x_i)], t_i) + \lambda_{\text{clip}} \cdot l_{\text{clip}}(\mathsf{I}_{\theta}(x_i) - \mathsf{T}_{\theta}(t_i)) \right]. \quad (2)$$

## 4.2 Downstream Image Difference Task

We consider downstream tasks where the goal is to describe differences between two visually similar images $x_1, x_2 \in \mathcal{X}$, satisfying $d_{\mathcal{X}}(x_1, x_2) \leq \epsilon$. The performance of a MLLM on this task can be evaluated using the generalization error:

$$G(\theta) = \mathbb{E}_{x_1, x_2 \sim \mathcal{P}(\mathcal{X}), \, t_1=f(x_1), \, t_2=f(x_2)} \left[ l_{\text{cap}} \left( \mathsf{Z}_{\theta}[\mathsf{I}_{\theta}(x_1) - \mathsf{I}_{\theta}(x_2)], \, \Delta_{\mathcal{T}}(t_1, t_2) \right) \right], \quad (3)$$

where $\mathsf{Z}_{\theta}[\mathsf{I}_{\theta}(x_1) - \mathsf{I}_{\theta}(x_2)]$ means that translating the feature-level image difference into a language-level description. Ideally, we would like the empirical minimizer $\hat{\theta} \in \arg \min \hat{R}(\theta)$ to achieve low generalization error, i.e., $G(\hat{\theta})$ is small. However, even state-of-the-art MLLMs (e.g., GPT-4o) still underperform on fine-grained difference description tasks. This raises a central question: *why does this performance gap persist?* There are two primary contributing factors. First, there may be *semantic under-specification*, where captions omit salient visual details, or *caption incompleteness*, where not all semantically meaningful aspects of an image $x$ are described in the corresponding caption $t$. More broadly, such cases fall under *visual-textual misalignment*, where the semantic content of the image and caption diverge. As these misalignments can range from mild to severe, we model them as noisy captions $\tilde{t} = \tilde{f}(x)$, where $\tilde{f}$ is an arbitrary, potentially biased captioning function. Second, even if the MLLM is trained on a clean dataset, a generalization gap can still arise when the model is trained to minimize $R(\theta)$ but evaluated on the downstream objective $G(\theta)$. For example, models in prior works [3, 21, 33, 55] were trained on datasets with rewritten captions. However, our experiments show that these models still perform poorly on tasks that require sensitivity to visual differences. The underlying intuition is that standard MLLMs trained using the loss in equation (2) lack the architectural capacity to explicitly recognize and describe fine-grained differences between similar images.

---

[4]CLIP loss is introduced for background information only. It is not used in our method's SFT stage.

| Model | Object | Attr. | Scene | Spatial | Action | Part | Count | Differ. | Compar. | Neg. | Univ. | Avg |
|---|---|---|---|---|---|---|---|---|---|---|---|---|
| Human | 85.71 | 100 | 100 | 100 | 100 | 93.75 | 100 | 75 | 100 | 100 | 92.86 | **95.21** |
| **API-based Models** | | | | | | | | | | | | |
| GPT-4o-2024-08-06 | 57.14 | 56.25 | 61.54 | 57.14 | 33.33 | 43.75 | 50.00 | **66.67** | 31.58 | 42.86 | **64.29** | 51.32 |
| GPT-4.1-2025-04-14 | **71.43** | **62.50** | 61.54 | **50.00** | 33.33 | **50.00** | **61.11** | **66.67** | 47.37 | 42.86 | 50.00 | **54.26** |
| Claude3.7 Sonnet | 57.14 | 43.75 | 46.15 | 35.71 | 46.67 | 25.00 | 38.89 | 50.00 | 15.79 | 42.86 | 42.86 | 40.44 |
| Qwen-VL-Plus-25-03 | 57.14 | 56.25 | 61.54 | 42.86 | 33.33 | 31.25 | 38.89 | **66.67** | 47.37 | **50.00** | 35.71 | 47.36 |
| Doubao-1.5-vision-pro | 69.23 | 37.50 | **69.23** | **50.00** | **50.00** | **50.00** | 38.89 | **66.67** | **52.63** | 42.86 | 42.86 | 51.81 |
| **Open-source Models** | | | | | | | | | | | | |
| Qwen2-VL-7B | 35.71 | 43.75 | **61.54** | 42.86 | 46.67 | 31.25 | 50.00 | 33.33 | 21.05 | 28.57 | 28.57 | 38.48 |
| Qwen2-VL-7B (ours) | 42.86 | 43.75 | 53.85 | **57.14** | **53.33** | 50.00 | **55.56** | **58.33** | 36.84 | 42.86 | 28.57 | 47.55 |
| Qwen2.5-VL-7B | 53.85 | 50.00 | 38.46 | 42.86 | 12.50 | 18.75 | 44.44 | 50.00 | 26.32 | 42.86 | 57.14 | 39.74 |
| Qwen2.5-VL-7B (ours) | 57.14 | 56.25 | 53.84 | 42.86 | 46.67 | 43.75 | **55.56** | 50.00 | **47.37** | **50.00** | **64.29** | **51.61** |
| LLaVA-V1.6-7B | **64.29** | 37.50 | 30.77 | 21.43 | 33.33 | 25.00 | 27.78 | 25.00 | 26.32 | 21.43 | 28.57 | 31.04 |
| LLaVA-V1.6-7B (ours) | 57.14 | 37.50 | 46.15 | 42.86 | 46.67 | 37.50 | 44.44 | 33.33 | 42.11 | 28.57 | 28.57 | 40.44 |
| LLaMA-3.2-11B | 46.15 | 43.75 | 38.46 | 50.00 | 50.00 | 25.00 | 22.22 | 33.33 | 15.79 | 21.43 | 35.71 | 34.71 |
| LLaMA-3.2-11B (ours) | 38.46 | **62.50** | 46.15 | 35.71 | 37.50 | 37.50 | 33.33 | 41.67 | 31.58 | 42.86 | 42.86 | 40.92 |

Table 1: **Model performance on the MED benchmark subtasks.** Each column corresponds to a specific type of task in MED benchmark: Object, Attr. (Attribute), Scene, Spatial, Action, Part, Count, Differ. (Differentiation), Compar. (Comparison), Neg. (Negation), Univ. (Universality), and Avg (overall accuracy). Bold values indicate the best performance per column within each model category. We use colors to highlight whether model trained with our method increases or decreases performance compared to the original model.

## 4.3 Supervised Fine-tuning

Motivated by the preceding two factors, we assume the MLLM $\texttt{M}_\theta$ is trained on a *noisy dataset* $\mathcal{D}_\eta$, where each image-caption pair $(x, t)$ satisfies that $t = f(x)$ with probability $\eta$, and $t = \tilde{f}(x)$ otherwise. That is, with probability $\eta$, the caption faithfully describes the image; with probability $1 - \eta$, it reflects an imperfect or incomplete description.

Accordingly, the baseline MLLM $\texttt{M}_\theta$ is trained on $\mathcal{D}_\eta$ by minimizing the empirical risk:

$$\hat{R}_\eta(\theta) = \frac{1}{|\mathcal{D}_\eta|} \sum_{(x_i, t_i) \in \mathcal{D}_\eta} \left[ \lambda_{\text{cap}} \cdot l_{\text{cap}}(\texttt{Z}_\theta[\texttt{I}_\theta(x_i)], t_i) + \lambda_{\text{clip}} \cdot l_{\text{clip}}(\texttt{I}_\theta(x_i) - \texttt{T}_\theta(t_i)) \right]. \tag{4}$$

This objective corresponds to the population-level risk:

$$R_\eta(\theta) = \eta \cdot R(\theta) + (1 - \eta) \cdot \tilde{R}(\theta), \tag{5}$$

where

$$\tilde{R}(\theta) = \mathbb{E}_{x \sim \mathcal{P}(\mathcal{X}),\ t = \tilde{f}(x)} \left[ \lambda_{\text{cap}} \cdot l_{\text{cap}}(\texttt{Z}_\theta[\texttt{I}_\theta(x)], t) + \lambda_{\text{clip}} \cdot l_{\text{clip}}(\texttt{I}_\theta(x) - \texttt{T}_\theta(t)) \right]. \tag{6}$$

Let $\hat{\theta}_\eta$ and $\theta_\eta^*$ denote the empirical and population risk minimizers under noisy supervision. Due to the presence of label noise and a mismatch between the training and evaluation objectives, the downstream generalization error $G(\hat{\theta}_\eta)$ may remain large. As a result, the model $\texttt{M}_{\hat{\theta}_\eta}$ exhibits suboptimal performance on fine-grained image difference tasks.

To mitigate this issue, we propose a supervised fine-tuning (SFT) approach using a curated dataset focused on image differences. Given a noisy dataset $\mathcal{D}_\eta$, for each pair $(x_i, \tilde{t}_i)$, we construct an augmented triplet as follows.

- Create the clean caption $t_i = f(x_i)$;
- Edit $\tilde{t}_i$ to obtain $\hat{t}_i$, and use an image generator (e.g., Gemini) to produce a visually similar image $\hat{x}_i$ such that $d_{\mathcal{X}}(x_i, \hat{x}_i) \leq \epsilon$ and $\hat{t}_i = f(\hat{x}_i)$;
- Use a prompting engine $\texttt{S}_\phi$ (e.g., an LLM) to generate a fine-grained natural language difference description $\texttt{S}_\phi(t_i, \hat{t}_i)$, where $\phi$ represents all parameters of the generator.

This process yields a curated dataset $\mathcal{D}_{\text{edit}}$ for post-training. We fine-tune the model $\texttt{M}_{\hat{\theta}_\eta^*}$ on this dataset by minimizing the following empirical SFT loss:

$$\hat{R}_{\text{SFT}}(\theta) = \frac{1}{|\mathcal{D}_{\text{edit}}|} \sum_{(x_i, \hat{x}_i, \texttt{S}_\phi(t_i, \hat{t}_i)) \in \mathcal{D}_{\text{edit}}} \left[ l_{\text{cap}} \left( \texttt{Z}_\theta[\texttt{I}_\theta(x_i) - \texttt{I}_\theta(\hat{x}_i)],\ \texttt{S}_\phi(t_i, \hat{t}_i) \right) \right]. \tag{7}$$

| Model | Pope | Coarse | Fine | Visual_Sim | Visual_Corr | Count | MMVP | Ave | MME |
|---|---|---|---|---|---|---|---|---|---|
| Qwen2-VL-7B | 92.50 | 71.21 | 48.24 | 51.11 | 30.23 | 55.83 | 31.33 | 54.35 | 1679.52 |
| Qwen2-VL-7B (Ours) | 96.27 | 73.92 | 46.16 | 51.85 | 33.72 | 59.17 | 32.67 | 56.25 | 1681.27 |
| Qwen2.5-VL-7B | 96.29 | 73.95 | 57.35 | 49.63 | 33.72 | 50.00 | 27.33 | 55.47 | 1685.14 |
| Qwen2.5-VL-7B (Ours) | 97.52 | 75.97 | 59.36 | 51.85 | 37.79 | 59.17 | 28.00 | 58.52 | 1701.87 |
| LLaVA-V1.6-7B | 95.56 | 58.28 | 31.93 | 51.11 | 21.51 | 45.83 | 28.67 | 47.56 | 1441.89 |
| LLaVA-V1.6-7B (Ours) | 97.39 | 56.74 | 35.13 | 48.14 | 24.42 | 49.17 | 30.00 | 48.71 | 1420.57 |
| LLaMA-3.2-11B | – | 69.03 | 48.94 | 43.70 | 20.93 | 44.17 | 26.00 | 42.13 | 1421.71 |
| LLaMA-3.2-11B (Ours) | – | 72.60 | 47.21 | 45.93 | 19.19 | 50.00 | 28.00 | 43.82 | 1430.67 |

Table 2: **Performance of models on other benchmarks.** Green indicates improved performance, and red indicates decreased performance after fine-tuning. Coarse and Fine correspond to the *Coarse Perception* and *Fine-grained Perception* sub-tasks in the MMStar; Visual_Sim, Visual_Corr, and Count correspond to the *Visual_Similarity*, *Visual_Correspondence*, and *Counting* sub-tasks in BLINK, respectively; Pope reports POPE precision, and MME is the MME perception score.

| Model | Pope | Coarse | Fine | Visual_Sim | Visual_Corr | Count | MMVP | Ave | MME |
|---|---|---|---|---|---|---|---|---|---|
| Qwen2-VL-7B | 92.50 | 71.21 | 48.24 | 51.11 | 30.23 | 55.83 | 31.33 | 54.35 | 1679.52 |
| Qwen2-VL-7B + trained-ViT | 93.79 | 73.81 | 49.17 | 51.11 | 31.40 | 53.28 | 32.00 | 54.94 | 1668.18 |
| Qwen2-VL-7B (Ours) | 96.27 | 73.92 | 46.16 | 51.85 | 33.72 | 59.17 | 32.67 | 56.25 | 1681.27 |
| Qwen2.5-VL-7B | 96.29 | 73.95 | 57.35 | 49.63 | 33.72 | 50.00 | 27.33 | 55.47 | 1685.14 |
| Qwen2.5-VL-7B + trained-ViT | 96.43 | 75.26 | 58.75 | 50.37 | 33.14 | 55.00 | 31.33 | 57.18 | 1694.83 |
| Qwen2.5-VL-7B (Ours) | 97.52 | 75.97 | 59.36 | 51.85 | 37.79 | 59.17 | 28.00 | 58.52 | 1701.87 |

Table 3: **Ablation results.** Green cells denote an improvement over the corresponding base model, red cells indicate a drop, and yellow marks no change. *All column abbreviations (Pope, Coarse, Fine, Visual_Sim, Visual_Corr, Count, MMVP, Ave, MME) carry the same meanings as in Table 2.*

By minimizing $\hat{R}_{\text{SFT}}(\theta)$, the model parameter is updated from the noisy initialization $\hat{\theta}_\eta$ to a refined estimate $\hat{\theta}_{\text{SFT}}$. The proposed loss function implicitly embeds an objective that forces the model to associate semantic image edits with structured changes in the feature space, and to maintain precise and consistent representations for unaltered image areas. Empirically, we find that the post-trained model $\text{M}_{\hat{\theta}_{\text{SFT}}}$ achieves significantly better performance on visual difference description tasks than the baseline $\text{M}_{\hat{\theta}_\eta}$. Moreover, the enhanced ability to reason about fine-grained visual changes also improves performance on classical downstream tasks, as it fosters deeper visual understanding and architectural awareness of subtle semantic shifts between images.

# 5 Experiments

We evaluate our approach's effectiveness in enhancing fine-grained visual difference comprehension in MLLMs. In Section 5.1, we outline our experimental settings. Section 5.2 presents findings from the Micro Edit Detection (MED) benchmark, designed to assess sensitivity to minor yet semantically significant visual differences. Section 5.3 examines generalization by testing our models across multiple multimodal benchmarks. Section 5.4 offers an ablation study analyzing how our training strategy's key design choices affect performance.

## 5.1 Experimental Settings

To assess our approach's effectiveness, we fine-tune several open-source vision-language models on the Micro Edit Dataset using our SFT loss (Equation 7). Models include Qwen2-VL-7B [55], Qwen2.5-VL-7B [3], LLaVA-V1.6-Vicuna-7B [33], and LLaMA-3.2-Vision-Instruct-11B [21]. All fine-tuning uses LLaMA-Factory [67] with consistent hyperparameters for fair comparison. We also evaluate recent commercial closed-source models as performance benchmarks: GPT-4o (2024-08-06) [26], GPT-4.1 (2025-04-14) [2], Claude 3.7 Sonnet [1], Qwen-VL-Plus (2025-03-18) [3], and Doubao-1.5-Vision-Pro (2025-03-28). Implementation details are shown in Appendix C.

## 5.2 Experimental Results on MED Benchmark

We evaluate our method on the Micro Edit Detection (MED) benchmark, which tests models' ability to detect and describe minimal yet semantically meaningful visual differences. Table 1 presents results across eleven task categories, comparing human performance, API-based models, and open-source models with and without our fine-tuning.

*Closed-source models still struggle with fine-grained differences understanding compare with human.* Human annotators achieve 95.21% accuracy, setting a high bar that current models struggle to reach. Even top closed-source models like GPT-4.1 (54.26%), Doubao-1.5-vision-pro (51.81%), and GPT-4o

(51.32%) fall short, particularly in relational tasks like Action, Comparison, and Negation—revealing persistent gaps in fine-grained visual reasoning.

*Our fine-tuning method significantly boosts open-source models.* Our supervised fine-tuning consistently enhances all tested open-source backbones. Qwen2-VL-7B improves from 38.48% to 47.55% (+9.07), with major gains in Spatial (+14.28), Action (+6.67), Count (+5.56), and Differentiation (+25.00). Qwen2.5-VL-7B increases from 40.24% to 51.61% (+11.37), excelling in Comparison (+21.06), Count (+22.23), and Universality (+14.29). LLaMA-3.2-11B rises from 34.71% to 40.92% (+6.21), and LLaVA-V1.6-7B from 40.44% to 44.04% (+3.60). These results demonstrate our approach's broad applicability across diverse architectures.

*Our best model outperforms several closed-source models on MED Benchmark.* Among open-source models, Qwen2.5-VL-7B (Ours) achieves the highest accuracy (51.61%), rivaling commercial models like GPT-4o (51.32%) and Doubao-1.5-Vision-Pro (51.81%). It also surpasses Claude 3.7 Sonnet (40.44%) and Qwen-VL-Plus (47.36%), despite its smaller scale and open-source nature. This demonstrates that with fine-grained data and alignment objectives, open-source vision-language models can match closed-source systems in tasks requiring precise visual difference understanding.

*Evaluation on real-world image pairs confirms generalization.* We evaluate all models on the MED-Real Set (50 minimally different real-world image pairs; see Section 3.3 and Appendix B) to assess generalization beyond synthetic edits. Fine-tuned models outperform their base versions: Qwen2.5-VL-7B improves from 66.0% to 74.0%, LLaVA-V1.6-7B from 38.0% to 50.0%, and LLaMA-3.2-11B from 36.00% to 50.00%. Qwen2-VL-7B improves from 78.0% to 80.0% accuracy, demonstrating strong intrinsic robustness. These results confirm effective generalization to real-world fine-grained visual differences.

## 5.3    Experimental Results on Other Benchmarks

To evaluate generalization beyond the MED benchmark, we test our fine-tuned models on several established multimodal benchmarks, including MMStar, BLINK, POPE, and MME shown in Table 2.

*Fine-tuning yields consistent and generalizable performance gains.* As shown in Table 2, across all models, our supervised fine-tuning improves both perception and reasoning. Qwen2.5-VL-7B (Ours) achieves the highest average score (58.52) and MME perception score (1701.87), outperforming all open-source baselines. Notable gains include Count (+9.17), Visual Correspondence (+4.07), and POPE (+1.23), reflecting stronger numeracy, spatial grounding, and hallucination resistance. These improvements hold across model families, confirming the effectiveness of fine-grained supervision.

*Robustness improvements reflected in Hallucination Benchmark.* Qwen2.5-VL-7B (Ours) achieves a POPE score of 97.52, ranking among the highest for open-source models, while LLaVA-V1.6-7B (Ours) improves from 95.56 to 97.39. This demonstrates stronger resistance to hallucinated object predictions, even under challenging prompts. Although minor decreases in Fine-grained Perception are observed—likely due to increased sensitivity to small edits—these are outweighed by overall improvements, confirming enhanced precision with minimal trade-offs.

## 5.4    Ablation Study

To analyze each component's contribution, we perform an ablation study with three settings: (1) base model without fine-tuning, (2) fine-tuning only the visual encoder (ViT) with the language model (LLM) frozen, and (3) joint fine-tuning of ViT and LLM (our full method). Table 3 presents the ablation results for Qwen2-VL-7B and Qwen2.5-VL-7B.

*Fine-tuning the vision encoder alone provides solid gains.* Fine-tuning the vision encoder (ViT) alone yields consistent improvements in perception-heavy tasks. For Qwen2.5-VL-7B, this results in a 1.31 gain in average score and nearly 10 on the MME perception score. Notable gains include Count (+5.00), Coarse Perception (+1.31), and MMVP (+4.00), highlighting that refining visual representations enhances low-level grounding and numeracy.

*Joint fine-tuning further boosts multimodal reasoning.* Joint fine-tuning of ViT and LLM achieves the best performance across metrics. For Qwen2.5-VL-7B, it improves the average score by +1.34 and MME by +7.04 compared to the vision-only model. Notable gains in Visual Correspondence (+4.65) and POPE (+1.09) underscore the importance of aligning the language head with updated visual features, which reduces hallucinations and enhances cross-modal integration.

# 6 Conclusions and Limitations

We propose a framework to improve fine-grained visual difference understanding in MLLMs by tackling two key challenges: limited semantically controlled data and weak alignment objectives. Our contributions include: (1) a scalable pipeline to generate minimally different image pairs, (2) the Micro Edit Dataset (MED) with 50K samples and a 215-item benchmark, and (3) a fine-tuning strategy with feature-level consistency to enhance robustness to small edits.

**Limitations and future directions.** Our framework focuses on binary edits between image pairs, with promising directions in extending it to multi-step transformations, temporal reasoning, and compositional edits. Full fine-tuning of vision and language components achieves strong performance, but exploring efficient adaptation methods could reduce computational costs. Applying our approach to unified-token architectures like Chameleon and Emu3 may further enhance its applicability.

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

# Appendices

# A  Examples of SFT Dataset

In this section, we provide some examples of our constructed MED dataset for supervised fine-tuning in Fig 4. It combines high-quality image pairs with corresponding questions and visually grounded answers, focusing on fine-grained reasoning tasks.

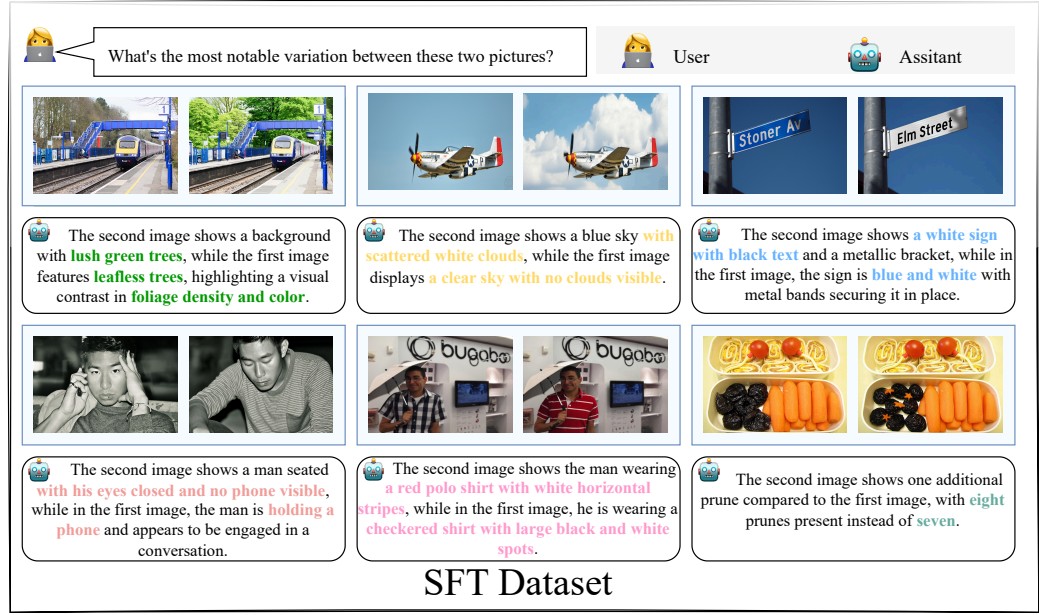

Figure 4: **Example samples from the Micro Edit Dataset (MED) used for supervised fine-tuning.** Each sample consists of a minimally edited image pair, a question prompting for the most notable difference, and a visually grounded answer. Answers are refined to highlight concrete semantic changes based on visual evidence. The full dataset contains over 50K such high-quality QA pairs supporting fine-grained multimodal reasoning.

# B  Experiments on MED-Real Set

We provide the evaluation results of the four models on the MED-Real set of our MED Benchmark in Tab 4. We evaluate all models on the MED-Real Set, consisting of 50 minimally different real-world image pairs, to assess their generalization beyond synthetic edits. Fine-tuned models show consistent improvements over their base versions: Qwen2.5-VL-7B improves from 66.00% to 74.00%, LLaVA-V1.6-7B from 38.00% to 50.00%, and LLaMA-3.2-11B from 36.00% to 50.00%. Qwen2-VL-7B from 78.00% to 80.00% . These findings highlight effective generalization to real-world, fine-grained visual differences.

| Model | Acc (%) |
|---|---|
| Qwen2-VL-7B | 78.00 |
| Qwen2-VL-7B (Ours) | 80.00 |
| Qwen2.5-VL-7B | 66.00 |
| Qwen2.5-VL-7B (Ours) | 74.00 |
| LLaVA-V1.6-vicuna-7B | 38.00 |
| LLaVA-V1.6-vicuna-7B (Ours) | 50.00 |
| Llama-3.2-11B-Vision-Instruct | 36.00 |
| Llama-3.2-11B-Vision-Instruct (Ours) | 50.00 |

Table 4: **Performance on MED-Real Set.** Green cells denote an improvement over the base model.

# C    Training Details

In this section, we present all the hyperparameters we used to training the three kinds of models in Table 5, Table 6 and Table 7. All the training processes were conducted using llamafactory [67]. Regarding image resolution and the number of image tokens, we adhere to the original settings specified by each model.

Table 5: **Hyperparameters for training Qwen2-VL & Qwen2.5-VL models**

| Hyperparameter | Value |
| --- | --- |
| LoRA Rank | 8 |
| LoRA $\alpha$ | 16 |
| LoRA Dropout | 0.1 |
| LoRA Target | all |
| GPU | $8 \times$ NVIDIA A800 |
| Batch Size | 16 |
| Gradient Accumulation Steps | 8 |
| Warmup Ratio | 0.1 |
| Learning Rate | 1e-4 |
| Learning Rate Scheduler | Cosine |
| Unfreeze Vision Tower | True |

Table 6: **Hyperparameters for training LLaVA-V1.6-7B model**

| Hyperparameter | Value |
| --- | --- |
| LoRA Rank | 8 |
| LoRA $\alpha$ | 16 |
| LoRA Dropout | 0.1 |
| LoRA Target | all |
| GPU | $8 \times$ NVIDIA A800 |
| Batch Size | 1 |
| Gradient Accumulation Steps | 8 |
| Warmup Ratio | 0.1 |
| Learning Rate | 1e-5 |
| Learning Rate Scheduler | Cosine |
| Unfreeze Vision Tower | False |

Table 7: **Hyperparameters for training LLaMA-3.2-11B model**

| Hyperparameter | Value |
| --- | --- |
| LoRA Rank | 8 |
| LoRA $\alpha$ | 16 |
| LoRA Dropout | 0.1 |
| LoRA Target | all |
| GPU | $8 \times$ NVIDIA A800 |
| Batch Size | 4 |
| Gradient Accumulation Steps | 4 |
| Warmup Ratio | 0.1 |
| Learning Rate | 1e-5 |
| Learning Rate Scheduler | Cosine |
| Unfreeze Vision Tower | True |

# D    More Examples of MED Benchmark

In this section, we present additional examples from the MED Benchmark in Fig 5 and Fig 6, including their types, answer options, correct answers, and the answers provided by GPT-4.1.

*How do these two images primarily differ from each other?*

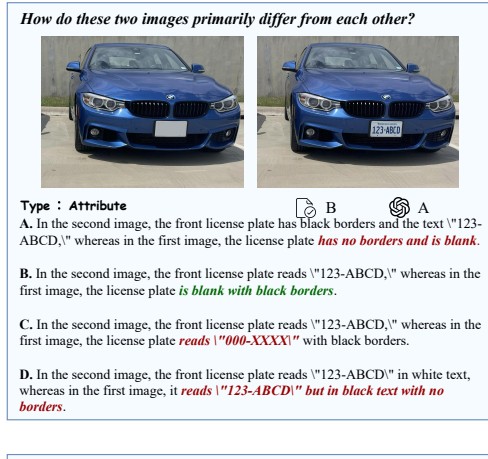

**Type : Attribute**  📄 B  🆖 A

**A.** In the second image, the front license plate has black borders and the text \"123-ABCD,\" whereas in the first image, the license plate *has no borders and is blank*.

**B.** In the second image, the front license plate reads \"123-ABCD,\" whereas in the first image, the license plate *is blank with black borders*.

**C.** In the second image, the front license plate reads \"123-ABCD,\" whereas in the first image, the license plate *reads \"000-XXXX\"* with black borders.

**D.** In the second image, the front license plate reads \"123-ABCD\" in white text, whereas in the first image, it *reads \"123-ABCD\" but in black text with no borders*.

---

*What's the most significant contrast between these two photos?*

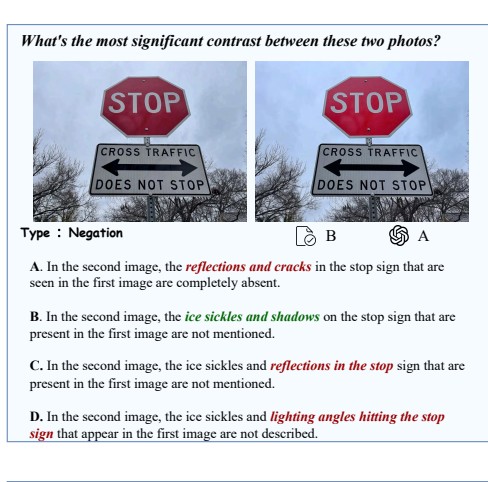

**Type : Negation**  📄 B  🆖 A

**A.** In the second image, the *reflections and cracks* in the stop sign that are seen in the first image are completely absent.

**B.** In the second image, the *ice sickles and shadows* on the stop sign that are present in the first image are not mentioned.

**C.** In the second image, the ice sickles and *reflections in the stop* sign that are present in the first image are not mentioned.

**D.** In the second image, the ice sickles and *lighting angles hitting the stop sign* that appear in the first image are not described.

---

*What distinguishes one image from the other?*

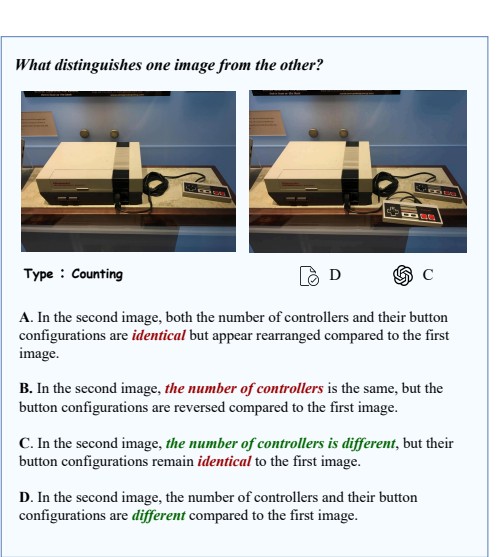

**Type : Counting**  📄 D  🆖 C

**A.** In the second image, both the number of controllers and their button configurations are *identical* but appear rearranged compared to the first image.

**B.** In the second image, *the number of controllers* is the same, but the button configurations are reversed compared to the first image.

**C.** In the second image, *the number of controllers is different*, but their button configurations remain *identical* to the first image.

**D.** In the second image, the number of controllers and their button configurations are *different* compared to the first image.

---

*How do these two images primarily differ from each other?*

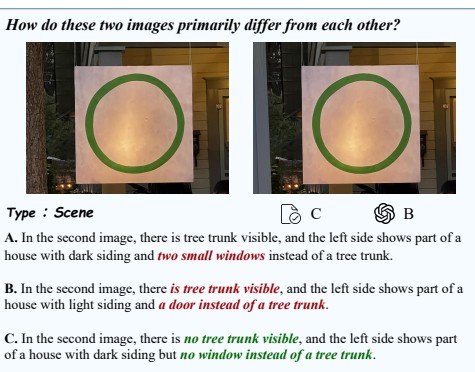

**Type : Scene**  📄 C  🆖 B

**A.** In the second image, there is tree trunk visible, and the left side shows part of a house with dark siding and *two small windows* instead of a tree trunk.

**B.** In the second image, there *is tree trunk visible*, and the left side shows part of a house with light siding and *a door instead of a tree trunk*.

**C.** In the second image, there is *no tree trunk visible*, and the left side shows part of a house with dark siding but *no window instead of a tree trunk*.

**D.** In the second image, there is no tree trunk visible, and the left side shows part of a house with dark siding and *a window* instead of a tree trunk.

---

*What's the most notable variation between these two pictures?*

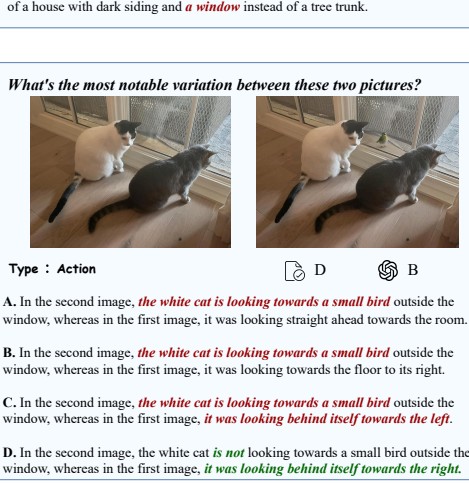

**Type : Action**  📄 D  🆖 B

**A.** In the second image, *the white cat is looking towards a small bird* outside the window, whereas in the first image, it was looking straight ahead towards the room.

**B.** In the second image, *the white cat is looking towards a small bird* outside the window, whereas in the first image, it was looking towards the floor to its right.

**C.** In the second image, *the white cat is looking towards a small bird* outside the window, whereas in the first image, *it was looking behind itself towards the left*.

**D.** In the second image, the white cat *is not* looking towards a small bird outside the window, whereas in the first image, *it was looking behind itself towards the right*.

---

*What is the fundamental difference you can see between these two visuals?*

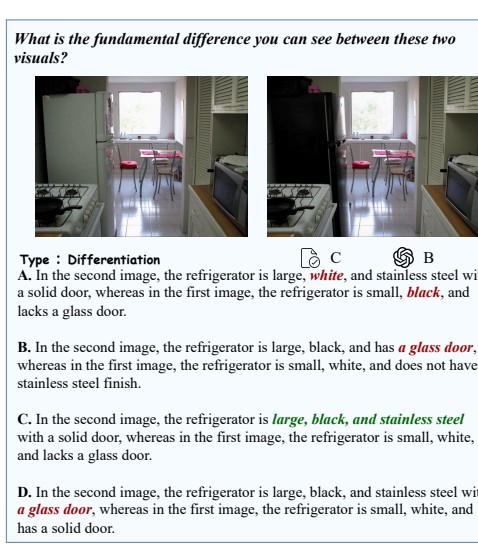

**Type : Differentiation**  📄 C  🆖 B

**A.** In the second image, the refrigerator is large, *white*, and stainless steel with a solid door, whereas in the first image, the refrigerator is small, *black*, and lacks a glass door.

**B.** In the second image, the refrigerator is large, black, and has *a glass door*, whereas in the first image, the refrigerator is small, white, and does not have a stainless steel finish.

**C.** In the second image, the refrigerator is *large, black, and stainless steel* with a solid door, whereas in the first image, the refrigerator is small, white, and lacks a glass door.

**D.** In the second image, the refrigerator is large, black, and stainless steel with *a glass door*, whereas in the first image, the refrigerator is small, white, and has a solid door.

---

Figure 5: More examples in the MED benchmark

none*What distinguishes one image from the other?*

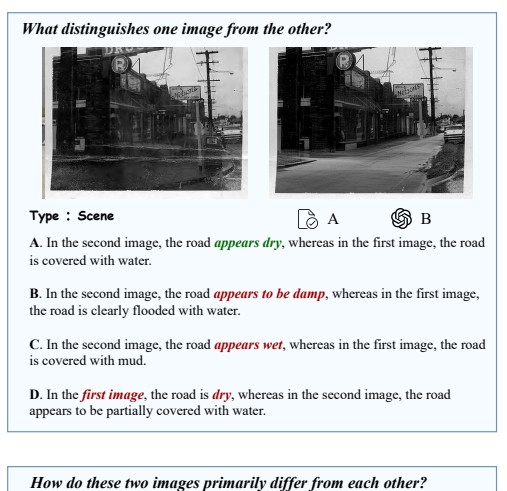

**Type : Scene**     A    B

**A**. In the second image, the road *appears dry*, whereas in the first image, the road is covered with water.

**B**. In the second image, the road *appears to be damp*, whereas in the first image, the road is clearly flooded with water.

**C**. In the second image, the road *appears wet*, whereas in the first image, the road is covered with mud.

**D**. In the *first image*, the road is *dry*, whereas in the second image, the road appears to be partially covered with water.

---

*How do these two images primarily differ from each other?*

*How do these two images primarily differ from each other?*

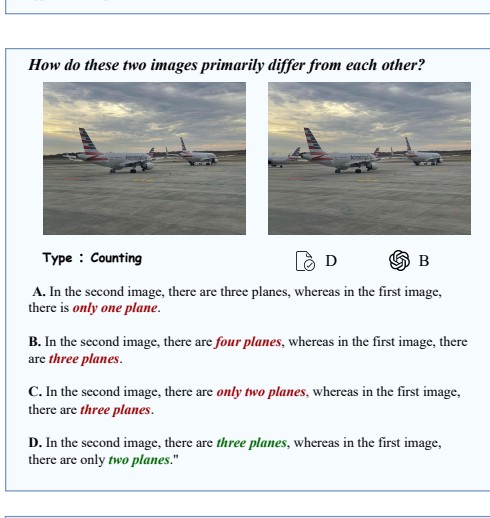

**Type : Counting**    D    B

**A.** In the second image, there are three planes, whereas in the first image, there is *only one plane*.

**B.** In the second image, there are *four planes*, whereas in the first image, there are *three planes*.

**C.** In the second image, there are *only two planes*, whereas in the first image, there are *three planes*.

**D.** In the second image, there are *three planes*, whereas in the first image, there are only *two planes*."

---

*What distinguishes one image from the other?*

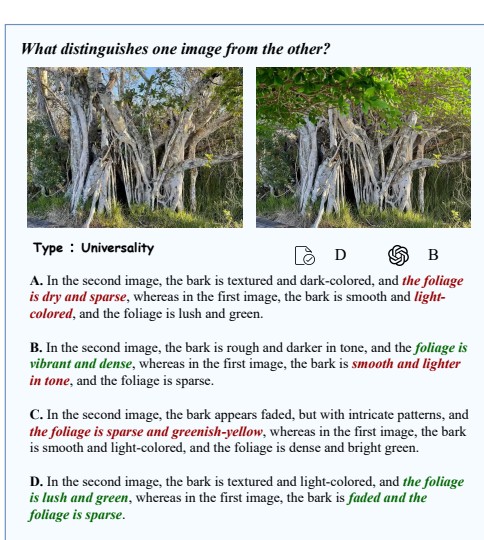

**Type : Universality**    D    B

**A.** In the second image, the bark is textured and dark-colored, and *the foliage is dry and sparse*, whereas in the first image, the bark is smooth and *light-colored*, and the foliage is lush and green.

**B.** In the second image, the bark is rough and darker in tone, and the *foliage is vibrant and dense*, whereas in the first image, the bark is *smooth and lighter in tone*, and the foliage is sparse.

**C.** In the second image, the bark appears faded, but with intricate patterns, and *the foliage is sparse and greenish-yellow*, whereas in the first image, the bark is smooth and light-colored, and the foliage is dense and bright green.

**D.** In the second image, the bark is textured and light-colored, and *the foliage is lush and green*, whereas in the first image, the bark is *faded and the foliage is sparse*.

---

*How do these two images primarily differ from each other?*

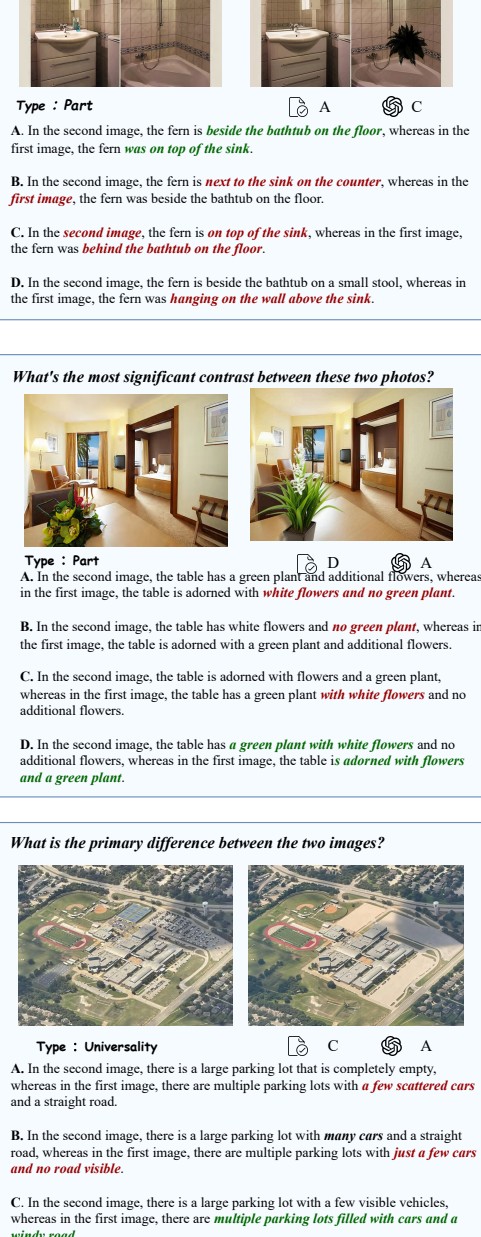

**Type : Part**    A    C

**A**. In the second image, the fern is *beside the bathtub on the floor*, whereas in the first image, the fern *was on top of the sink*.

**B**. In the second image, the fern is *next to the sink on the counter*, whereas in the *first image*, the fern was beside the bathtub on the floor.

**C**. In the *second image*, the fern is *on top of the sink*, whereas in the first image, the fern was *behind the bathtub on the floor*.

**D**. In the second image, the fern is beside the bathtub on a small stool, whereas in the first image, the fern was *hanging on the wall above the sink*.

---

*What's the most significant contrast between these two photos?*

**Type : Part**    D    A

**A.** In the second image, the table has a green plant and additional flowers, whereas in the first image, the table is adorned with *white flowers and no green plant*.

**B.** In the second image, the table has white flowers and *no green plant*, whereas in the first image, the table is adorned with a green plant and additional flowers.

**C.** In the second image, the table is adorned with flowers and a green plant, whereas in the first image, the table has a green plant *with white flowers* and no additional flowers.

**D.** In the second image, the table has *a green plant with white flowers* and no additional flowers, whereas in the first image, the table is *adorned with flowers and a green plant*.

---

*What is the primary difference between the two images?*

**Type : Universality**    C    A

**A.** In the second image, there is a large parking lot that is completely empty, whereas in the first image, there are multiple parking lots with *a few scattered cars* and a straight road.

**B.** In the second image, there is a large parking lot with *many cars* and a straight road, whereas in the first image, there are multiple parking lots with *just a few cars and no road visible*.

**C.** In the second image, there is a large parking lot with a few visible vehicles, whereas in the first image, there are *multiple parking lots filled with cars and a windy road*.

**D.** In the second image, there are *multiple parking lots filled with cars* and a windy road, whereas in the first image, there is a single large parking lot with a few visible vehicles.

Figure 6: More examples in the MED benchmark

# E  Prompts

In this section, we systematically present all the prompts used in the construction of the Micro Edit Dataset. These prompts are designed to cover every stage of the dataset creation process, including filtering editable images, generating editing instructions, producing detailed descriptions for both original and edited images, describing the differences between image pairs, conducting evaluation, and generating training as well as evaluation data.

Specifically:

- Figure 7 shows the prompt for filtering editable images, ensuring that the selected images are suitable for subsequent editing tasks.

- Figure 8 illustrates the prompt for generating image editing instructions, which enables the creation of diverse editing scenarios.

- Figures 9 and 10 demonstrate the prompts for generating comprehensive descriptions of the original and the edited images, respectively, providing essential context for later analysis.

- Figure 11 presents the prompt for describing the differences between two images, facilitating both quantitative and qualitative analysis of micro edits.

- Figures 12 and 13 display the prompts used for evaluating the quality of the image edits, ensuring an objective assessment process.

- Figure 14 shows the prompt for generating Supervised Fine-Tuning (SFT) data, which supplies high-quality training samples for model fine-tuning.

- Figure 15 provides the question template used for benchmark evaluations, enabling a systematic assessment of model performance on micro editing tasks.

- Figures 16 and 17 demonstrate the prompts for generating the correct and distractor answers, respectively, enhancing the scientific rigor and difficulty of the benchmark.

All prompts have been carefully crafted and iteratively refined to ensure high quality and diversity in the dataset, laying a solid foundation for subsequent research and algorithm development on micro editing tasks.

"""
Please evaluate the following sample--which includes an image and accompanying text (question-answer pair or description)--based on the criteria listed below. Your response must STRICTLY follow the format of the provided example. For each criterion, provide a brief analysis, and conclude with a clear "Yes" or "No" recommendation.

Caption of the image: {caption}
---

## **Evaluation**

**Final Recommendation:** **Yes** or **No**

**1. Image_quality:**
- **Clarity:** [Evaluate if the image is sharp, free of blur, and low in noise]
- **Main_Subject:** [Evaluate if there is a clearly identifiable primary subject in the image]
- **Background_Complexity:** [Evaluate whether the background is simple and free of excessive clutter]

**2. Text_Clarity (Description):**
- **Specifity:** [Evaluate if the description is clear, specific, and unambiguous]
- **Uniquenes:** [Evaluate if the text provides straightforward, unique information without ambiguity]

**3. Editing_Potential:**
- **Identifiable_Editable_Elements:** [Evaluate if the image contains clearly identifiable objects or attributes that can be modified]
- **Impact_on_Text_Accuracy:** [Evaluate whether applying an edit would make the original text incorrect]

**4. Interference_Factors:**
- **Visual_Distractions:** [Evaluate whether there are distracting elements or complex scenes that might hinder a clear edit]
- **Subjectivity_in_Text:** [Evaluate if the text contains subjective aspects that could interfere with evaluation]

**Explanation:** [Provide a brief explanation supporting your recommendation]

---

IMPORTANT: Your answer MUST follow this EXACT format, with bold section headers and bullet points as shown above. Your Final Recommendation must clearly state either "**Yes**" or "**No**" in bold.
"""

Figure 7: Prompt for Qwen-2.5-VL-72B to filter editable images

```
"""
You are a professional image prompt engineer specializing in image editing instructions.

Analyze the provided image and any accompanying text, then generate a clear editing prompt based on what you see.

Description of the image: {}

Your editing prompt must follow this exact structure:
"Hi, This is a photo of [brief description of the current image]. Can you [specific editing instruction] and output the image?"

Choose an appropriate editing instruction from these composition categories:

BASIC COMPOSITIONS:
- Object: Adding or removing entities (people, animals, food, vehicles, objects)
- Attribute: Changing visual properties (color, material, size, shape)
- Scene: Modifying backgrounds or settings
- Spatial Relation: Altering physical arrangements (right, left, on top, below, inside)
- Action Relation: Changing interactions between entities
- Part Relation: Modifying part-whole relationships (body parts, clothing, accessories)

ADVANCED COMPOSITIONS:
- Counting: Adjusting quantity of entities
- Differentiation: Distinguishing objects by attributes
- Comparison: Changing relative characteristics
- Negation: Removing elements
- Universality: Applying changes to all members of a group

Examples:
- "Hi, This is a photo of a dog with biscuits on its head. Can you subtract one biscuit on its head and output the image?"
- "Hi, This is a picture of a highway. Can you add two vehicles on its left side and output the image?"
- "Hi, This is a photo of a table. Can you only move the phone out of this scene and output the image?"

Based solely on what you observe in the provided image, first indicate which composition type you're using (Prioritize
COMPOSITIONS other than 'Object'. Only choose 'Object' if other types cannot effectively demonstrate the editing possibilities
of the image.), then create the appropriate editing prompt.
Your response should be in this format:
## Type: [composition type]\n## Prompt: [your editing prompt]\n
"""
```

Figure 8: Prompt for Qwen-2.5-VL-72B to generate editing prompt

```
prompt_template =
"""Given an image, its current caption, and an edit prompt, your task is to determine whether the caption fully incorporates all information
from the edit prompt.
If not, you need to update the caption to include any missing information.

Image: [The image will be provided]
Current Caption: {original_description}
Edit Prompt: {edit_prompt}

IMPORTANT: The complete caption should ONLY describe what is ACTUALLY VISIBLE in the image. DO NOT add information from
the edit prompt if those elements don't actually exist in the image.

Follow these steps:
1. Carefully examine the image and understand the current caption
2. Identify all new or modified information requested in the edit prompt
3. Verify if each element from the edit prompt is actually visible in the image
4. Check whether the current caption already contains all relevant and visible information from the edit prompt
5. If the caption FULLY incorporates all relevant information from the edit prompt, respond with:
   "Caption is complete. No updates needed."
6. If the caption is MISSING any relevant information that is ACTUALLY VISIBLE in the image, create an updated caption by:
   - Preserving relevant parts of the original caption
   - Naturally integrating ONLY the missing information that is actually visible in the image
   - Maintaining a coherent, fluent, and accurate description
   - Ensuring the updated caption accurately describes what is visible in the image
   - EXCLUDING any information from the edit prompt that is not actually present in the image

Provide only the updated caption without any explanations, or confirm that no updates are needed (By Answer exactly "NO UPDATE").
"""
```

Figure 9: Prompt for Qwen-2.5-VL-72B to generate complete original description for original images

"
Your task is generating an accurate caption for the given image, using the following information as context:

Reference description: {description}
Key aspect to focus on: {type} ({type_description})

Your task is to:
1. Carefully examine the actual image content
2. Note any differences between the image and the reference description, especially regarding the {type} aspects
3. Create an accurate caption that accurately describes ONLY what is truly visible in the image

IMPORTANT:
- Your description must be based solely on what is actually visible in the image
- Do not include details from the reference description that aren't present in the image
- Pay special attention to accurately describing the {type} elements
- Be precise and objective in your description

Please provide your accurate caption in English, beginning with "CAPTION:".
"

Figure 10: Prompt for Qwen-2.5-VL-72B to generate description for edited images

"""Given these two image captions and a specific type of difference:

Caption 1: {complete_caption}
Caption 2: {new_caption}

Type class:
- Object: Adding or removing entities (people, animals, food, vehicles, objects)
- Attribute: Changing visual properties (color, material, size, shape)
- Scene: Modifying backgrounds or settings
- Spatial Relation: Altering physical arrangements (right, left, on top, below, inside)
- Action Relation: Changing interactions between entities
- Part Relation: Modifying part-whole relationships (body parts, clothing, accessories)
- Counting: Adjusting quantity of entities
- Differentiation: Distinguishing objects by attributes
- Comparison: Changing relative characteristics
- Negation: Removing elements
- Universality: Applying changes to all members of a group

Identify only the most obvious differences between these captions and choose a Type class the difference belong to.

Respond with just the key differences, using one sentence and one type class
Please strctly follow this format:"TYPE:XXX\nDIFFERENCE:XXXX"
"""

Figure 11: Prompt for Qwen-2.5-VL-72B to generate difference description

"""Determine if the described difference matches the edit prompt:

Edit Prompt: {edit_prompt}
Described Difference: {difference_description}

Assess whether the described difference correctly reflects what was requested in the edit prompt.

Answer with 1. Only "Yes" if they match. No explanation needed.
2. Only "No" if there's no obvious change. No explanation needed.
"""

Figure 12: Prompt for Qwen-2.5-VL-72B to get judge

```
"""
Your task is determine if the 'Described Difference' matches the 'Second Description':

Second Description: {second_edit_prompt}
Described Difference: {difference_description}

IMPORTANT GUIDELINES FOR EVALUATION:

- "Caption 1" refers to the first image description, and "Caption 2" refers to the second image description.
- FOCUS ON MEANING, NOT EXACT WORDING: Synonyms, paraphrasing, and alternative expressions that convey the same
meaning should be considered equivalent.
- ALLOW FOR VARIATION: The difference description may use different phrasing or perspective than the second description while
still being correct.
- ACCEPT STRUCTURAL DIFFERENCES: Different sentence structures, word order, or narrative approaches that communicate the
same content are valid.
- RECOGNIZE CONCEPTUAL MATCHES: When categories, concepts, or observations are semantically similar (even if using
different terminology), treat them as matches.
- EVALUATE BASED ON CORE CHANGES: If both descriptions identify the same fundamental changes between images, even
through different expressions, consider them aligned.
- BE GENEROUS IN INTERPRETATION: When in doubt, lean toward accepting descriptions that capture the essential elements, even
if details vary.

Answer with 1. Only "Yes" if they match. No explanation needed.
2. Only "No" if there's no obvious change. No explanation needed.
"""
```

Figure 13: Prompt for Qwen-2.5-VL-72B to get second judge

```
"""
I need your help to rewrite a description of differences between two images.

Current description (needs improvement):
"{diff_description}"

Please rewrite this description to focus on the VISUAL DIFFERENCES between the two images, NOT just differences
in captions.

Important guidelines:
- Assume "Caption 1" refers to the first image, and "Caption 2" refers to the second image
- Your rewritten description should describe what appears in the second image compared to the first image
- Use formats like "The second image shows..." or "In the first image, ... while in the second image..."
- Focus on actual visual elements that are present or absent in each image
- Be specific about what was added, removed, or changed
- Keep your description concise but informative

Your response must follow this exact format:
IMPROVED_DESCRIPTION: [Your rewritten description here]

Do not include any other text, explanations, or formatting in your response.
"""
"""
```

Figure 14: Prompt for Qwen3-32B to generate SFT data

```
# Question template
variations = [
"What is the primary difference between the two images?",
"What is the main distinction between these two images?",
"What's the key difference between these two pictures?",
"Can you identify the principal disparity between these two images?",
"How do these two images primarily differ from each other?",
"What's the most significant contrast between these two photos?",
"What distinguishes one image from the other?",
"What is the fundamental difference you can see between these two visuals?",
"What's the most notable variation between these two pictures?"
]
```

Figure 15: Question Template of Benchmark

```
"""
You are analyzing the differences between two images described by captions. Your task is to generate a clear answer that describes how
the second image differs from the first image.

Input: A difference description that compares elements between two images.
Output: A precise statement that explains how the second image (Caption 2) differs from the first image (Caption 1).

Your output MUST follow this exact format:
ANSWER: [Your description of how the second image differs from the first image]

Example:
Input difference description: "The deer on the far right is grazing with his head facing toward the grass in the first caption, while in the
second caption, it stands alone facing slightly to the right."
Output:
ANSWER: In the second image, the deer on the far right is standing alone facing slightly to the right, whereas in the first image, the deer
was grazing with its head facing toward the grass.

Important requirements:
1. Always frame your answer to describe how the second image differs from the first image
2. Use precise language to describe the specific differences
3. Begin with "In the second image" to maintain clarity
4. Always follow the ANSWER: format exactly as shown
"""
```

Figure 16: Prompt for GPT-4o to generate right answer

```
"""Please create exactly 3 completely incorrect statements about the difference between two images that are similar to the correct answer
but wrong in subtle ways.

Context:
- The first image (Caption 1) and second image (Caption 2) show the same scene with a specific difference
- The actual difference is: "{difference_description}"
- The correct statement that describes this difference is: "{right_answer}"

Your task:
1. Create exactly 3 statements that are deceptively similar to the correct answer but entirely incorrect
2. Each wrong answer should be closely related to the actual difference, but contain crucial errors
3. Make the wrong answers similar enough to the right answer that they could confuse someone
4. The statements should sound plausible and require careful attention to distinguish from the correct answer
5. Each wrong answer should suggest a specific difference between "the first image" and "the second image"
6. Use similar wording, structure, and level of detail as the correct answer

Output format:
```
wrong_answer1: [Your first wrong answer here]
wrong_answer2: [Your second wrong answer here]
wrong_answer3: [Your third wrong answer here]
```

Remember, your statements should be subtly wrong in ways that make them challenging to distinguish from the correct answer!"""
```

Figure 17: Prompt for GPT-4o to generate wrong answer

# F  Analysis on Text Diversity

To further validate the quality and diversity of the generated instructions in our dataset, we computed
standard lexical diversity metrics. Specifically, we report the Moving Average Type-Token Ratio
(MATTR) with a window size of 50 for both our proposed dataset and the LLaVA-Instruct-150k
dataset for comparison.

Our dataset achieves a MATTR score of **0.7393** and LLaVA-Instruct-150k scores **0.7834**. These
quantitative results demonstrate that our instructions exhibit rich lexical diversity and low redundancy
in phrasing, which is a notable characteristic for a Visual Question Answering (VQA) dataset.

# G  Additional Analysis on CLIP-Score Threshold Selection

To determine the optimal CLIP-score threshold for constructing our MED dataset, we conducted an
ablation study evaluating the trade-off between data quality and scale. This analysis complements
our approach for MED-Bench, where we follow [54] in using a 95% similarity threshold.

We trained identical Qwen2.5-VL-7B models on MED subsets filtered using different CLIP-similarity thresholds ($\delta$), with comprehensive evaluation across 12 reasoning categories. As shown in Table G, our analysis reveals several key findings:

Higher thresholds ($\delta \geq 0.8$) produced better per-sample alignment, with $\delta = 0.8$ achieving 78.6% accuracy on universal tasks. However, this came at the cost of significantly reduced data size (36k samples), leading to performance degradation in compositional reasoning tasks such as spatial reasoning (-21.2% versus $\delta = 0.7$) and part (-12.5%).

Conversely, thresholds below $\delta < 0.7$ introduced noisy image-text pairs that degraded performance across all categories.

The optimal threshold of $\delta = 0.7$ with 50k samples achieved the highest average accuracy (51.5%) by balancing two critical factors: (1) preserving sufficient diversity for complex reasoning tasks, evidenced by an 18.8% improvement in attribute binding compared to $\delta = 0.8$; and (2) maintaining scale advantages, as demonstrated by the 10.3% performance gap between the full 50k dataset and a random 12k subset at the same threshold.

This systematic analysis confirms that $\delta = 0.7$ optimally balances data quality and quantity for robust model training.

Table 8: Performance comparison across different CLIP-similarity thresholds ($\delta$)

| Model ($\delta$) | Data Size | Avg | Object | Attr. | Scene | Spatial | Action | Part | Count | Differ. | Compar. | Neg. | Univ. |
|---|---|---|---|---|---|---|---|---|---|---|---|---|---|
| $\delta = 0.7$ | 50k | **51.51** | 57.14 | 56.25 | 53.84 | 42.86 | 46.67 | 43.75 | 55.56 | 50.00 | 47.37 | 50.00 | 64.29 |
| $\delta = 0.7$ | random-12k | 41.21 | 53.85 | 43.75 | 53.85 | 42.86 | 62.50 | 18.75 | 22.22 | 33.33 | 31.58 | 50.00 | 50.00 |
| $\delta = 0.8$ | 36k | 46.67 | 53.84 | 37.50 | 46.15 | 21.43 | 43.75 | 31.25 | 61.11 | 50.00 | 42.11 | 50.00 | 78.57 |
| $\delta = 0.8$ | random-12k | 42.42 | 61.54 | 31.25 | 53.85 | 28.57 | 37.50 | 37.50 | 33.33 | 41.67 | 36.84 | 64.86 | 50.00 |
| $\delta = 0.9$ | 12k | 44.85 | 46.15 | 50.00 | 46.15 | 35.71 | 43.75 | 43.75 | 38.89 | 50.00 | 31.58 | 57.14 | 57.14 |

# H   Extended Evaluation on BLIP Model Family

To further demonstrate the broad applicability of our approach across different model architectures, we extended our MED benchmark evaluation to include the BLIP series, specifically the newest BLIP-3o model. This evaluation covers all 11 semantic edit types and compares performance both before and after fine-tuning on our dataset.

As shown in Table 9, After fine-tuning with our method, BLIP-3o-8B demonstrates improved performance on the MED benchmark, with average accuracy increasing from 44.24% to 50.90%. These results validate that our fine-tuning approach generalizes effectively beyond the Qwen and LLaVA families, providing consistent performance improvements across diverse model architectures and establishing the broad applicability of our methodology.

Table 9: Performance comparison of BLIP-3o-8B on MED benchmark before and after fine-tuning

| Model | Avg | Obj | Attr | Scene | Spatial | Action | Part | Count | Differ | Compar | Neg | Univ |
|---|---|---|---|---|---|---|---|---|---|---|---|---|
| BLIP-3o-8B | 44.24 | 46.15 | 50.00 | 69.23 | 50.00 | 37.50 | 31.25 | 44.44 | 41.67 | 31.58 | 50.00 | 42.86 |
| BLIP-3o-8B (Ours) | **50.90** | 38.46 | **62.50** | 46.15 | 42.86 | **62.50** | 37.50 | **66.67** | 50.00 | **52.63** | 35.71 | **57.14** |

# I   Backbone Scaling Ablation Study

To evaluate the general applicability of our approach across different model scales, we conducted an ablation study by fine-tuning Qwen2.5-VL models of varying sizes (3B and 7B parameters) on the same MED dataset. The performance results, measured across 12 reasoning categories, are presented in Table 10.

Table 10: Performance comparison of Qwen2.5-VL models at different scales before and after fine-tuning on MED

| Model | Avg | Obj | Attr | Scene | Spatial | Action | Part | Count | Differ | Compar | Neg | Univ |
|---|---|---|---|---|---|---|---|---|---|---|---|---|
| Qwen2.5-VL-3B | 33.94 | 38.46 | 37.50 | 53.85 | 21.43 | 25.00 | 43.75 | 27.78 | 16.67 | 26.32 | 35.71 | 50.00 |
| Qwen2.5-VL-3B (Ours) | 38.79 | 46.15 | 43.75 | 38.46 | 35.71 | 43.75 | 43.75 | 38.89 | 16.67 | 42.11 | 28.57 | 42.86 |
| Qwen2.5-VL-7B | 39.74 | 53.85 | 50.00 | 38.46 | 42.86 | 12.50 | 18.75 | 44.44 | 50.00 | 26.32 | 42.86 | 57.14 |
| Qwen2.5-VL-7B (Ours) | 51.61 | 57.14 | 56.25 | 53.84 | 42.86 | 46.67 | 43.75 | 55.56 | 50.00 | 47.37 | 50.00 | 64.29 |

Our experimental results demonstrate that both model variants benefit substantially from our supervised fine-tuning strategy. The 3B parameter model shows an average accuracy improvement from 33.94% to 38.79%, while the 7B parameter model achieves a more pronounced gain from 39.74% to 51.61%. These consistent improvements across different model scales confirm that the effectiveness of our MED approach is robust and scalable, independent of the backbone model size. This ablation study validates our core design choice and highlights the general applicability of our method.

## J  Efficiency of LoRA Fine-tuning

In our experiments, we employed LoRA (Low-Rank Adaptation) for all models due to its favorable efficiency-performance trade-off. As summarized in Table 11, LoRA significantly reduces training time while achieving performance comparable to full-parameter fine-tuning across different model architectures.

Table 11: Comparison of LoRA and full-parameter fine-tuning

| Model | Fine-tuning Method | Training Time (50k data) |
|---|---|---|
| Qwen2.5-VL-7B | LoRA | 6 hours |
| | Full | 29 hours |
| LLaVA-1.6-7B | LoRA | 13 hours |
| | Full | 57 hours |
| LLaMA3.2-11B-VL | LoRA | 8 hours |
| | Full | 70+ hours |

These results indicate that for fine-grained visual understanding tasks, the necessary model adaptations can be effectively achieved through low-rank updates without requiring full retraining of the model backbone. This suggests that such tasks primarily require adjustments in higher-layer representations rather than fundamental changes to the base model parameters.

## K  Analysis of Image Quality Filtering and Metric Selection

To ensure the quality and semantic consistency of edited images in our MED dataset, we implemented a comprehensive image filtering pipeline. Our primary approach adopts CLIP-based embedding similarity, following the methodology established by [54], to maintain high-level semantic alignment between edited images and their originals.

We complemented this CLIP-based analysis with Structural Similarity Index (SSIM) measurements to evaluate low-level visual consistency across our 50K image pairs. The mean SSIM score is 0.5011 (std: 0.2028, max: -0.0134, min 0.9839) confirm that most edits introduce structurally subtle modifications while preserving overall image integrity, aligning with our objective of fine-grained semantic editing.

Our selection of CLIP over SSIM as the primary filtering metric was based on several considerations. SSIM is known to struggle with localized edits on uniform backgrounds [17], whereas CLIP demonstrates superior robustness to superficial visual changes while focusing on semantic-level alignment. This characteristic makes CLIP particularly suitable for our task of ensuring conceptual consistency in fine-grained image editing.

For instruction-image alignment validation, we employed a dual approach: instructions were directly used as generation inputs to the Gemini model, followed by manual verification on over 1,000 samples. This comprehensive validation ensures strong semantic alignment between textual instructions and their corresponding visual edits through both procedural enforcement and human evaluation.

## L  Broader Impact

This work aims to improve fine-grained visual understanding in multimodal large language models (MLLMs), which can positively impact applications requiring visual precision, such as robotics, assistive technologies, and industrial inspection. However, enhanced sensitivity to subtle visual differences could also increase risks of misuse, such as generating more convincing disinformation or

enabling deceptive visual narratives. While our dataset is based on synthetic edits over public images and does not introduce direct privacy concerns, models fine-tuned on it may still inherit underlying biases from pretraining. We recommend responsible deployment, especially in safety-critical settings, and see our benchmark as a tool to help identify and mitigate hallucination risks in future systems.

