# OpenReview forum: "Hallucination at a Glance: Controlled Visual Edits and Fine-Grained Multimodal Learning"
_NeurIPS.cc/2025/Conference — NeurIPS 2025 poster_

### Official Review · Reviewer_K4PJ · 2025-06-10

[review text omitted: it was posted to a different submission]

---

> ### Author Rebuttal · Authors · 2025-07-31
>
> **Overall Response**
>
> Thanks for the thoughtful and constructive feedback! In response, we expanded the MED-Real set to 50 real-world samples, and included results on BLIP-3o to demonstrate architectural generality. We also clarified our hallucination mitigation strategy, feature-level supervision design, and benchmark construction process. We will include all clarifications and results in the revised version.
>
> > **Response to W1: Scale of MED-Real and Generalization to Real-World Benchmarks**
> >
>
> We appreciate the reviewer’s suggestion to strengthen claims of real-world generalizability with more robust evaluation.
>
> **1. Expanded MED-Real evaluation**: While the original MED-Real set contained 35 real-world image pairs, we have now expanded it to 50 examples by adding 15 newly curated pairs. The table below shows the accuracy progression with increasing real-image samples:
>
> | Model with sft scalability |    acc（15） |  acc（35） | acc（50） |
> | --- | --- | --- | --- |
> | Qwen2-VL-7B | 66.67 | 82.86 | 78.00 |
> | Qwen2-VL-7B(ours) | 73.33 | 82.86 | 80.00 |
> | Qwen2.5-VL-3B | 46.67 | 54.29 | 52.00 |
> | Qwen2.5-VL-3B(ours) | 53.33 | 57.14 | 56.00 |
> | Qwen2.5-VL-7B | 60.00 | 68.57 | 66.00 |
> | Qwen2.5-VL-7B(ours) | 73.33 | 74.29 | 74.00 |
> | LLaVA-1.6-7B | 46.67 | 34.29 | 38.00 |
> | LLaVA-1.6-7B(ours) | 60.00 | 45.71 | 50.00 |
> | LLaMA3.2-11B-VL | 26.67 | 40.00 | 36.00 |
> | LLaMA3.2-11B-VL(ours) | 46.67 | 51.43 | 50.00 |
>
> Across all model families, MED-finetuning results in clear performance gains. The improvements are consistent as the real-world sample set grows, which suggests that the model is not simply overfitting to synthetic distribution patterns.
>
> **2. General benchmark validation (Table 3, Page 8)**: To further validate generalizability, we report results on a range of diverse real-image benchmarks, including POPE, MMStar, BLINK, and MME. These datasets contain non-synthetic, real-world image distributions and diverse tasks. For example:
>
> - Qwen2.5-VL-7B (Ours) improves:
>     - POPE: 96.29 → 97.52
>     - MME: 1685.14 → 1701.87
>     - BLINK Visual Corr: 33.72 → 37.79
>
> These results show that MED improves real-world performance even on unrelated tasks, indicating robust, transferable fine-grained reasoning capabilities — not simply domain overfitting.
>
> We will include this clarification and the expanded table in the revision to emphasize MED’s impact on generalizable multimodal understanding.
>
> > **Response to W2: Hallucination Risk from Qwen-VL and Gemini**
> >
>
> We thank the reviewer for raising the important concern about potential hallucinations from Qwen2.5-VL-72B and Gemini in our data generation pipeline. We fully agree that vision-language models are not immune to hallucination, which is why our pipeline is explicitly designed to minimize and isolate such errors through two key strategies: task decomposition and multi-stage filtering.
>
> **1. Task decomposition: minimizing hallucination through controlled prompts**: Instead of relying on the model to generate free-form captions or edits, we break the data generation into small, grounded subtasks:
>
> - Qwen2.5-VL is only asked to revise or extend existing captions, based on concrete visual prompts (e.g., “add a red ball”).
> - Edited captions are generated with strong contextual guidance, including the original caption and the specified edit type.
> - Difference descriptions are produced by a separate text-only model (Qwen3-32B) to avoid compounding visual noise.
>
> This decomposition ensures that each model step is *tightly scoped and grounded in the image*, significantly reducing hallucination risks.
>
> **2. Multi-stage filtering: detecting and removing faulty samples**: To further control quality, we apply multiple filtering layers:
>
> - Qwen2.5-VL first evaluates raw image-caption pairs via a structured quality rubric, discarding low-alignment or low-editability samples.
> - After editing, we use CLIP-based similarity filtering to ensure visual differences are minimal and semantically plausible.
> - Finally, we perform manual verification on 1,000 image pairs, covering all 11 edit types, to identify and eliminate rare hallucination patterns and refine prompt templates. According to the manual verification results, we designed heuristic rules to filter out low-quality data.
>
> Through this process, we ensure that both Qwen-VL and Gemini are used only within narrow, controllable contexts, with human- and CLIP-based safeguards in place. We will add clarification and provide statistics (e.g., instruction rejection rates) in the revised version to further support the robustness of our pipeline.
>
> > **Response to W3: Meta-Text Artifacts and Benchmarking Noise**
> >
>
> We thank the reviewer for raising the concern about meta-text phrases such as *“in the second caption”* or *“is described”*, which could introduce subtle benchmarking noise or false negatives.
>
> We clarify that such phrasing appears only in intermediate textual difference descriptions generated during Step 3 of our data construction pipeline (see Section 3.2).
>
> Before creating the final MED benchmark, we perform a systematic transformation to ensure questions are fully visually grounded:
>
> - All contrastive difference sentences are rewritten into QA-style prompts that refer exclusively to visual content, not to how things are described in text.
> - Each QA item includes four answer choices, and we perform manual verification to ensure:
>     - Only one answer is visually correct.
>     - Distractors are visually plausible but wrong.
>     - No options rely on or reference the structure of captions or linguistic cues like “caption” or “mention”.
>
> This transformation ensures that no meta-textual language appears in the final benchmark, and that model evaluation remains image-grounded. We will make this clarification explicit in the final version and include an illustrative before/after example to demonstrate how these artifacts are resolved.
>
> > **Response to W4: Broader Evaluation Across Model Architectures**
> >
>
> We thank the reviewer for highlighting the importance of evaluating across diverse model families.
>
> While our primary evaluations focused on Qwen, LLaVA, and LLaMA-style vision-language models, we emphasize that our method is model-agnostic and readily applicable to other architectures. To demonstrate this, we additionally applied our fine-grained supervision method to BLIP-3o, a strong open-source model architecturally distinct from the LLaVA/Qwen family.
>
> | Model | avg | obj | attr | scene | spatial | action | part | count | differ | compar | neg | univ |
> | --- | --- | --- | --- | --- | --- | --- | --- | --- | --- | --- | --- | --- |
> | BLIP-3o-8B | 44.24 | 46.15 | 50.00 | 69.23 | 50.00 | 37.50 | 31.25 | 44.44 | 41.67 | 31.58 | 50.00 | 42.86 |
> | BLIP-3o-8B(ours) | 50.90 | 38.46 | 62.50 | 46.15 | 42.86 | 62.50 | 37.50 | 66.67 | 50.00 | 52.63 | 35.71 | 57.14 |
>
> The finetuned BLIP-3o model shows significant improvements across many dimensions, especially in *attribute*, *action*, *counting*, and *difference* reasoning. This supports our claim that the proposed fine-grained visual edit supervision enhances a model’s reasoning ability regardless of its backbone architecture.
>
> In total, we evaluate and improve models from four diverse model families:
>
> - Qwen series (VL-7B / VL-2.5-7B / 3B)
> - LLaVA series (1.6-Vicuna)
> - LLaMA-Vision (3.2-11B)
> - BLIP-3o (Transformer with Querying Transformer head)
>
> We will include this extended evaluation in the revised version to reinforce the generality and adaptability of our approach across architectures.
>
> > **Response to W5: Image Quality and Clarity**
> >
>
> Thank you for the valuable feedback. In the revised version, we will provide higher-resolution images and add reference captions to enhance clarity and interpretability.
>
> > **Response to Q1: Feature Consistency Loss and Hallucination Reduction**
> >
>
> Thank you for the insightful question. Our feature consistency loss reduces hallucinations by improving visual representations prior to decoding, not by modifying the autoregressive decoding itself.
>
> Specifically, the loss encourages stable and semantically aligned embeddings for minimally edited image pairs, which enhances the fidelity of the input to the language decoder. This grounding helps reduce hallucinations during text generation by ensuring that the decoder is conditioned on more reliable visual features.
>
> As shown in Table 3, even updating only the vision encoder leads to improved performance on perception-heavy tasks (e.g., Count, Coarse), highlighting that better visual representations alone significantly reduce errors. Full gains are achieved with joint tuning, but the primary hallucination mitigation stems from the representation level, not decoding.
>
> In summary, the feature consistency loss acts at the representation level, ensuring the visual embeddings are robust and well-grounded. This improved representation serves as a more reliable foundation for the standard autoregressive decoder, which then produces more accurate and less hallucinatory text as a downstream effect. We hope this clarifies the interaction. We thank the reviewer again for the valuable feedback.

---

> > ### Comment · Reviewer_K4PJ · 2025-08-09
> >
> > I appreciate the authors for addressing my concerns; I have no additional questions about the paper.

---

> > > ### Author Response · Authors · 2025-08-09
> > > **Thanks for your reply and positive assessment!**
> > >
> > > Thank you for your time and valuable feedback! We sincerely appreciate your effort in reviewing our work and are grateful for your positive assessment. We will carefully incorporate your insights to further refine the final manuscript.

---

> ### Author Response · Authors · 2025-08-05
> **Kind Reminder: Your Additional Thoughts on Paper #9063 Are Welcome**
>
> Dear Reviewer  K4PJ,
>
> Your expert feedback is crucial to refining this work. While we fully understand the discussion period may pose challenges for your schedule, we would value the chance to clarify any final points with you prior to its conclusion on Aug 8.
>
> We hope we've been able to address your questions and concerns so far. We would be glad to address any further concerns you may have, and we will try our best to clarify promptly.
>
> Thank you again for your feedback and comments; they were really helpful!
>
> Warm Regards, Authors of Submission #9063

---

### Official Review · Reviewer_PV47 · 2025-06-29

**Clarity:** 3
**Significance:** 3
**Originality:** 3
**Rating:** 4
**Confidence:** 5

**Summary:**

The paper points out that current MLLMs struggles at fine-grained visual differences, which may further lead to inaccurate predictions or hallucinations. To address this, the authors designed an automated data generation pipeline, yielding the Micro Edit Dataset (MED). This dataset comprises over 50K image-text pairs across 11 fine-grained edit categories, specifically designed to highlight subtle visual changes.  Then, the author validate the effectiveness of the dataset with different MLLMs like Qwen 2.5-VL. The results demonstrate that the dataset not only enhances MLLM performance on the dedicated MED benchmark but also improves performance on other VL tasks.

**Questions:**

1. I am curious about the scalability of the Micro Edit Dataset (MED). If the dataset size were continuously scaled up, would performance improvements persist?
2. The authors employed LoRA for fine-tuning the model. What would be the impact of not using LoRA? Additionally, the hyper-parameters vary significantly across different base models. What might explain this discrepancy?
3.  I noticed that CLIP loss is incorporated into all presented equations. However, most existing MLLMs do not typically integrate CLIP loss during SFT. Is CLIP loss indeed utilized in the paper's SFT process?
4. The paper claims to use a feature-level consistency loss. How is this implemented within the proposed method? If I understand correctly, the SFT loss only requires the model to distinguish differences between two input images.

**Ethical Concerns:**

["NO or VERY MINOR ethics concerns only"]

**Final Justification:**

After careful consideration, I will maintain my score as "borderline accept".

**Limitations:**

yes

**Quality:**

3

**Strengths And Weaknesses:**

# Strengths
1. The identification of existing MLLMs' weakness in recognizing subtle visual changes is significant. Current SOTA MLLMs remain considerably behind human-level recognition abilities (e.g., GPT-4.1-2025-04-14 at 54% versus human performance at 95%), highlighting a crucial direction for future MLLM optimization.
2. The dataset generation pipeline is well-designed, incorporating various filtering and editing techniques to ensure the production of a high-quality dataset.
3. The authors rigorously validated the MED across diverse MLLMs, utilizing both the MED benchmark and other VL benchmarks. This extensive validation confirms the dataset's utility in enhancing MLLM recognition capabilities.

# Weakness
1. Some details are not clear. See Questions.

---

> ### Author Rebuttal · Authors · 2025-07-31
>
> **Overall Response**
>
> Thanks so much for the time and thoughtful feedback! Based on these comments, we conducted detailed scalability experiments and efficiency comparisons between LoRA and full fine-tuning. We also clarified the role of CLIP loss, hyperparameter settings, and the implicit feature-level consistency enforced by our SFT objective. These updates confirm the scalability, efficiency, and soundness of our method, and we will incorporate them into the final version if the paper is accepted.
>
> > **Response to Q1: Dataset Scalability of MED**
> >
>
> We thank the reviewer for the valuable question regarding the scalability of the Micro Edit Dataset (MED).
>
> To assess this, we trained models on progressively larger subsets of MED, ranging from 10k to 50k samples. The results are shown in the table below:
>
> | Model with sft scalability | avg | obj | attr | scene | spatial | action | part | count | differ | compar | neg | univ |
> | --- | --- | --- | --- | --- | --- | --- | --- | --- | --- | --- | --- | --- |
> | 10k | 40.00 | 53.85 | 25.00 | 46.15 | 57.14 | 18.75 | 31.25 | 44.44 | 58.33 | 36.84 | 50.00 | 28.57 |
> | 20k | 42.42 | 53.85 | 25.00 | 53.85 | 42.86 | 37.50 | 50.00 | 50.00 | 50.00 | 31.58 | 35.71 | 42.86 |
> | 30k | 43.64 | 46.15 | 56.25 | 61.54 | 61.54 | 37.50 | 31.25 | 27.78 | 33.33 | 33.33 | 35.71 | 50.00 |
> | 40k | 47.88 | 53.85 | 56.25 | 38.46 | 50.00 | 50.00 | 43.75 | 50.00 | 50.00 | 36.84 | 42.86 | 57.14 |
> | 50k | **51.51** | 57.14 | 56.25 | 53.84 | 42.86 | 46.67 | 43.75 | 55.56 | 50.00 | 47.37 | 50.00 | 64.29 |
>
> The results show a consistent improvement in performance as the dataset size increases. The model trained with the full 50k MED samples achieves the highest average accuracy and shows stronger capability in nearly all categories. This demonstrates that MED supports scalable learning and that its fine-grained design continues to provide useful supervision as more data is added.
>
> > **Response to Q2: Impact of Using LoRA and Hyperparameter Differences**
> >
>
> We thank the reviewer for the thoughtful question regarding the use of LoRA and the variation in hyperparameters across base models.
>
> **1. LoRA vs. full-parameter fine-tuning**: In our experiments, we chose to apply LoRA for all models due to its strong efficiency-performance trade-off. As shown below, LoRA not only significantly reduces training time, but also achieves comparable—or in most cases, nearly identical—performance to full fine-tuning:
>
> | **Model** | **Fine-tuning Method** | **Time (50k data)** | **Performance Impact** |
> | --- | --- | --- | --- |
> | Qwen2.5-VL-7B | LoRA | 6 hours | Similar |
> |  | Full | 29 hours | – |
> | LLaVA-1.6-7B | LoRA | 13 hours | Similar |
> |  | Full | 57 hours | – |
> | LLaMA3.2-11B-VL | LoRA | 8 hours | Similar |
> |  | Full | 70+ hours | – |
>
> These results suggest that for fine-grained visual understanding, most of the necessary capability can be adapted through low-rank adaptation. This may be because such tasks primarily require adjustments in higher-layer representations, rather than full re-training of the model backbone.
>
> **2. Hyperparameter variations across models**: Regarding the hyperparameter differences across base models, the variations largely reflect architecture-specific tuning practices. Our implementation follows the LLaMA-Factory framework, and the learning rate, batch size, and warmup ratios we use are based on commonly adopted settings within that community for each model family. The variation arises from differences in:
>
> - Model scale (e.g., 3B vs 7B vs 11B),
> - Tokenization behavior and sequence lengths,
> - Pretraining objectives and convergence speeds.
>
> We will clarify this in the appendix and include the exact configurations for each model.
>
> > **Response to Q3: Role of CLIP Loss vs. SFT Loss**
> >
>
> We thank the reviewer for the close reading and for raising this important point.
>
> To clarify: although our paper discusses CLIP loss (Equations 2 and 4) as part of the general pretraining framework for MLLMs, we do not use the CLIP loss during the supervised fine-tuning (SFT) stage proposed in our method.
>
> The presentation of the CLIP-style loss is meant to formally contextualize the limitations of existing training objectives (e.g., Eq. 2, 4, and 6) and to motivate the need for a task-aligned loss for difference reasoning. As explained in Section 4.2 and 4.3, models trained under these conventional objectives struggle with downstream fine-grained difference detection, due to both noise in supervision and objective mismatch.
>
> Our proposed SFT method addresses this by replacing the standard objective with a targeted difference-aware training loss. The actual SFT objective used in our method is shown in Equation (7):
>
> $$\hat{R}\_{\text{SFT}}(\theta) = \frac{1}{|\mathcal{D}\_{\text{edit}}|} \sum\_{(x\_i, \hat{x}\_i, \texttt{S}\_{\phi}(t\_i, \hat{t}\_i)) \in \mathcal{D}\_{\text{edit}}} \left[
> l\_{\text{cap}}\left(\texttt{Z}\\_{\theta}[\texttt{I}\_{\theta}(x\_i) - \texttt{I}\_{\theta}(\hat{x}\_i)],\ \texttt{S}\_{\phi}(t\_i, \hat{t}\_i)\right)
> \right].$$
>
> This loss does not include any CLIP-style component, and is purely designed to supervise the model to describe visual differences between two images in a contrastive, semantically grounded way.
>
> We will revise the text to make it more explicit that CLIP loss is discussed only to frame prior training regimes and is not used in our final supervised fine-tuning pipeline.
>
> > **Response to Q4: Feature-Level Consistency Loss Implementation**
> >
>
> We thank the reviewer for the careful reading of our manuscript and for raising this insightful question. We appreciate the opportunity to clarify the role of the *feature-level consistency* in our proposed method.
>
> Your understanding is correct—the supervised fine-tuning (SFT) loss requires the model to distinguish and describe the differences between two input images. Importantly, the so-called *feature-level consistency loss* is not introduced as an explicit auxiliary loss term, but rather is implicitly embedded in the design of the SFT loss itself.
>
> As shown in Equation (7) of our paper, the loss is defined as:
>
> $$\hat{R}\_{\text{SFT}}(\theta) = \frac{1}{|\mathcal{D}\_{\text{edit}}|} \sum\_{(x\_i, \hat{x}\_i, \texttt{S}\_{\phi}(t\_i, \hat{t}\_i)) \in \mathcal{D}\_{\text{edit}}} \left[
> l\_{\text{cap}}\left(\texttt{Z}\_{\theta}[\texttt{I}\_{\theta}(x\_i) - \texttt{I}\_{\theta}(\hat{x}\_i)],\ \texttt{S}\_{\phi}(t\_i, \hat{t}\_i)\right)
> \right].$$
>
> Here:
> - $I\_{\theta}(x\_i)$ and $I\_{\theta}(\hat{x}\_i)$ denote the feature-level embeddings of the original and edited images, respectively, produced by the image encoder $I\_{\theta}$.
> - Their difference, $I_{\theta}(x_i) - I_{\theta}(\hat{x}_i)$, is passed to the text decoder $Z\_{\theta}$, which is trained to generate the correct description $y\_i$ of the visual change.
>
> This design implicitly enforces feature-level consistency: for the model to succeed in generating a correct difference description, the encoder must produce embeddings where fine-grained, semantically meaningful edits in the image space correspond to predictable and structured changes in feature space.
>
> Thus, the model is encouraged to:
>
> - Produce *stable visual representations* for unedited content.
> - Encode *fine-grained differences* in a way that can be directly mapped to textual explanations.
>
> This implicit regularization improves the model’s fine-grained reasoning ability and reduces hallucination, as shown in our experimental results across both synthetic and real benchmarks. We will clarify this point more explicitly in the revised version.

---

> > ### Comment · Reviewer_PV47 · 2025-08-09
> >
> > Thanks for providing such informative response. It has addressed my concerns. Therefore, I will keep my rating as borderline accept.

---

> > > ### Author Response · Authors · 2025-08-09
> > > **Thanks for your reply and positive assessment!**
> > >
> > > Thank you for your time and valuable feedback! We sincerely appreciate your effort in reviewing our work and are grateful for your positive assessment. We will carefully incorporate your insights to further refine the final manuscript.

---

> ### Author Response · Authors · 2025-08-05
> **Kind Reminder: Your Additional Thoughts on Paper #9063 Are Welcome**
>
> Dear Reviewer  PV47,
>
> Your expert feedback is crucial to refining this work. While we fully understand the discussion period may pose challenges for your schedule, we would value the chance to clarify any final points with you prior to its conclusion on Aug 8.
>
> We hope we've been able to address your questions and concerns so far. We would be glad to address any further concerns you may have, and we will try our best to clarify promptly.
>
> Thank you again for your feedback and comments; they were really helpful!
>
> Warm Regards, Authors of Submission #9063

---

### Official Review · Reviewer_ysLX · 2025-07-03

**Clarity:** 3
**Significance:** 3
**Originality:** 2
**Rating:** 4
**Confidence:** 4

**Summary:**

This paper tackles hallucinations in large VLMs, a problem of high interest in LM research. The authors hypothesize the root of hallucinations of large VLMs is inadequate training data requiring fine-grained visual understanding. The authors propose the Micro Edit Dataset, which provides both a training and benchmarking portion for improving and evaluating performance on fine-grained visual differences. The authors perform extensive fine-tuning on their dataset based on risk minimization over hallucination. They show that the SFT procedure results in higher performance across a number of standard VLM benchmarks.

**Questions:**

Can the authors comment on why performance seems to be a large margin higher on the MED-Real set vs the main MED benchmark? Are there any particular reasons stemming from the candidate data distribution, or is this an artifact of the model training domain? In this case, how can it be verified that the performance gains on fine-tuning on the MED were not primarily due to bridging the domain gap between real and synthetic?

Can the authors comment on the quite frequent artifacts of the texts referring to samples as texts? Will this not introduce false negatives in benchmarked models?

Can the authors demonstrate qualitative or quantitative ablations over the data curation pipeline, showing why specific similarity thresholding, etc. necessary to curate the best dataset?

Overall I have a positive impression of this work, but addressing my questions would encourage me to improve my rating. Thanks!

**Ethical Concerns:**

["NO or VERY MINOR ethics concerns only"]

**Final Justification:**

I maintain my positive review. I maintain concern about some of my initially raised points, as there were many text artifacts presented in the paper, and the revised data was not shared for qualitative evaluation. Additionally, while the expansion of real-image evaluation is appreciated, 50 images is still not a very large sample size. I recognize these concerns are difficult to address during a rebuttal period, and I remain positive overall about the contribution and core idea of the work.

**Limitations:**

Yes.

**Paper Formatting Concerns:**

None.

**Quality:**

3

**Strengths And Weaknesses:**

Strengths:
The problem is clearly of high importance, and the visual quality of paired images seems high. While not totally realistic visually, the resulting performance gains seem clear to validate the pipeline to be built upon as parts of larger VLM fine-tuning pipelines. The formulation of SFT as empirical risk minimization for hallucination makes sense, and results in nice performance gains.

Weaknesses:
There are artifacts of the text-based difference caption generation in multiple of the presented examples. For instance, the texts include phrases like "are not mentioned," "is described," "in the second caption." This raises questions about potential false negatives during benchmarking due to selecting the next best choice that describes an image instead of a caption.

As the dataset is a main contribution of the paper, more extensive qualification behind design choices (e.g. similarity thresholding, failure cases, ablations on backbone LLMs) would help strengthen the usefulness of the MED.

Some of the images are qualitatively obviously synthetic. The performance on the MED-Real set is much higher than analogous performance on the synthetic set (admittedly the real set is small).

---

> ### Author Rebuttal · Authors · 2025-07-30
>
> **Overall Response**
>
> Thanks for the insightful feedback! Based on thse comments, we conducted new ablation studies (on similarity threshold and model scale), expanded the MED-Real set, and clarified benchmark construction procedures to ensure robustness and generalization. We address all concerns in detail below and will incorporate these updates into the final version if the paper is accepted.
>
> > **Response to W1: Text Artifacts in Difference Descriptions and Benchmark Reliability**
> >
>
> We thank the reviewer for pointing out the phrasing artifacts such as *“is described”*, *“are not mentioned”*, or *“in the second caption”*, which appear in some of our difference descriptions.
>
> We want to clarify that these phrases originally appeared in intermediate contrastive descriptions produced during Step 3 of our data construction pipeline (Section 3.2). However, before finalizing the benchmark question-answer format, we apply a systematic transformation of these descriptions into visually grounded natural language questions. Specifically:
>
> - We rewrite contrastive captions into QA-style prompts, phrased purely in terms of *visual content*, not textual description.
> - All questions in the MED benchmark are then presented with four answer choices, and we manually verify every question-answer set (Section 3.3) to ensure that:
>     - Exactly one option is visually correct.
>     - The other options are plausible but visually incorrect.
>     - No options rely on meta-knowledge of captions or textual artifacts.
>
> This human verification step is essential to prevent the kind of false negatives the reviewer is concerned about — e.g., cases where a model might “correctly” describe the image but get penalized due to linguistic mismatches in the caption text.
>
> We will add clarification and examples in the final version showing how we resolve these caption artifacts during the transition from raw difference sentences to benchmark QA pairs, ensuring evaluation is image-grounded and artifact-free.
>
> > **Response to W2: Design Choices and Ablations in MED Construction and Q3: Impact of Similarity Threshold and Backbone Scaling**
> >
>
> We thank the reviewers for emphasizing the importance of justifying our dataset design, especially regarding similarity thresholding and backbone model selection. We have now included **two additional sets of ablation studies** to strengthen this point:
>
> **1. Similarity threshold (δ) ablation**: We compare model performance when training on subsets of MED with different CLIP-based similarity thresholds (δ). Results show a clear trend: stricter thresholds (e.g., δ = 0.8 or 0.9) yield better per-sample quality, but smaller data sizes limit overall performance. Meanwhile, δ = 0.7 with full 50k samples yields the best average accuracy, balancing quality and data scale.
>
> | Model with different δ | Data size | avg | obj | attr | scene | spatial | action | part | count | differ | compar | neg | univ |
> | --- | --- | --- | --- | --- | --- | --- | --- | --- | --- | --- | --- | --- | --- |
> | δ=0.7 | 50k | **51.51** | 57.14 | 56.25 | 53.84 | 42.86 | 46.67 | 43.75 | 55.56 | 50.00 | 47.37 | 50.00 | 64.29 |
> | δ=0.7 | random-12k | 41.21 | 53.85 | 43.75 | 53.85 | 42.86 | 62.50 | 18.75 | 22.22 | 33.33 | 31.58 | 50.00 | 50.00 |
> | δ=0.8 | 36k | 46.67 | 53.84 | 37.50 | 46.15 | 21.43 | 43.75 | 31.25 | 61.11 | 50.00 | 42.11 | 50.00 | 78.57 |
> | δ=0.8 | random-12k | 42.42 | 61.54 | 31.25 | 53.85 | 28.57 | 37.50 | 37.50 | 33.33 | 41.67 | 36.84 | 64.86 | 50.00 |
> | δ=0.9 | 12k | 44.85 | 46.15 | 50.00 | 46.15 | 35.71 | 43.75 | 43.75 | 38.89 | 50.00 | 31.58 | 57.14 | 57.14 |
>
> **2. Backbone scaling ablation**: To test the general applicability across model scales, we fine-tuned Qwen2.5-VL-3B and 7B on the same data. As shown below, both 3B and 7B variants benefit significantly from our SFT strategy, demonstrating that MED’s effectiveness is robust across different backbone sizes.
>
> | **Model** | **Avg** | **Obj** | **Attr** | **Scene** | **Spatial** | **Action** | **Part** | **Count** | **Differ** | **Compar** | **Neg** | **Univ** |
> | --- | --- | --- | --- | --- | --- | --- | --- | --- | --- | --- | --- | --- |
> | Qwen2.5-VL-3B | 33.94 | 38.46 | 37.50 | 53.85 | 21.43 | 25.00 | 43.75 | 27.78 | 16.67 | 26.32 | 35.71 | 50.00 |
> | Qwen2.5-VL-3B (Ours) | **38.79** | 46.15 | 43.75 | 38.46 | 35.71 | 43.75 | 43.75 | 38.89 | 16.67 | 42.11 | 28.57 | 42.86 |
> | Qwen2.5-VL-7B | 39.74 | 53.85 | 50.00 | 38.46 | 42.86 | 12.50 | 18.75 | 44.44 | 50.00 | 26.32 | 42.86 | 57.14 |
> | Qwen2.5-VL-7B (Ours) | **51.61** | 57.14 | 56.25 | 53.84 | 42.86 | 46.67 | 43.75 | 55.56 | 50.00 | 47.37 | 50.00 | 64.29 |
>
> Together, these ablations validate the core design choices behind MED—highlighting how both threshold calibration and model scalability are accounted for. We will include these new results and discussion in the revised version.
>
> > **Response to W3: Synthetic Visual Quality and Real Set Performance Gap**
> >
>
> We thank the reviewer for pointing out the visual quality difference between synthetic images and the MED-Real subset. While some synthetic samples may appear artificial upon close inspection, we ensured semantic fidelity through CLIP-based filtering with a cosine similarity threshold > 0.95. The average CLIP similarity gap between synthetic and real pairs is < 5%, suggesting the visual differences are minimal in embedding space.
>
> As shown below, the performance gap between MED and MED-Real is not uniformly large, especially for LLaVA and LLaMA models:
>
> | Model | MED (Avg) | MED-Real (Acc %) |
> | --- | --- | --- |
> | Qwen2-VL-7B | 38.48 | 82.86 |
> | Qwen2-VL-7B (Ours) | 47.55 | 82.86 |
> | Qwen2.5-VL-7B | 39.74 | 68.57 |
> | Qwen2.5-VL-7B (Ours) | 51.61 | 74.29 |
> | LLaVA-V1.6-vicuna-7B | 31.04 | 34.29 |
> | LLaVA-V1.6-vicuna-7B (Ours) | 40.44 | 45.71 |
> | LLaMA-3.2-11B-Vision-Instruct | 34.71 | 40.00 |
> | LLaMA-3.2-11B (Ours) | 40.92 | 51.43 |
>
> These results suggest that our synthetic dataset generalizes well to real-world data. The high real-set performance of Qwen models may be partially due to pretraining exposure to Visual Genome–style content, which is consistent with the source of MED-Real. However, for other models like LLaVA and LLaMA, the real/synthetic performance is comparable, indicating no evidence of domain overfitting
>
> > **Response to Q1: MED vs. MED-Real Performance and Domain Concerns**
> >
>
> We appreciate the reviewer’s question on the observed performance gap between the synthetic MED benchmark and the MED-Real set. We address this from three perspectives:
>
> 1. **Performance gap is not uniformly large**: As we noted in our response to W3, while Qwen models show higher performance on MED-Real (possibly due to pretraining exposure), models like LLaVA and LLaMA exhibit only minor differences between MED and MED-Real. This suggests that the synthetic nature of MED is not the primary limitation for those models.
>
> 2. **Improvement is not just from bridging synthetic–real gap**: Our fine-tuned models improve not only on MED, but also on **unrelated, real-world general benchmarks** that do not involve synthetic data. As shown in **Table 2 (Page 8) for full results**:
>
>     | Model | Ave | MME |
>     | --- | --- | --- |
>     | Qwen2-VL-7B | 54.35 | 1679.52 |
>     | Qwen2-VL-7B (Ours) | 56.25 | 1681.27 |
>     | Qwen2.5-VL-7B | 55.47 | 1685.14 |
>     | Qwen2.5-VL-7B (Ours) | 58.52 | 1701.87 |
>     | LLaVA-V1.6-7B | 47.56 | 1441.89 |
>     | LLaVA-V1.6-7B (Ours) | 48.71 | 1420.57 |
>     | LLaMA-3.2-11B | 42.13 | 1421.71 |
>     | LLaMA-3.2-11B (Ours) | 43.82 | 1430.67 |
>
>     These results confirm that our method improves fine-grained perception capabilities broadly, rather than simply adapting models to synthetic artifacts.
>
> 3. **Careful design to avoid domain artifacts**: To further avoid domain leakage:
>
>     - The MED-Real set is strictly held out from any fine-tuning.
>     - Our instruction pipeline includes both open-domain instructions and domain-agnostic edit types.
>     - We apply manual verification to ensure semantic correctness and diversity in both real and synthetic settings.
>
> Together, these points support that our performance gains reflect true improvements in fine-grained reasoning, not merely domain adaptation. We will clarify this in the final version.
>
> > **Response to Q2: Benchmark Robustness to Text-Based Artifacts**
> >
>
> We understand the reviewer’s concern that textual phrasing in the dataset might cause false negatives—for example, if a model selects the most accurate visual description, but the correct answer is tied to wording about the captions themselves.
>
> We emphasize that such risks are fully mitigated through two key measures:
>
> 1. All benchmark questions are rewritten to remove references to text or caption structure.
> 2. Manual verification ensures that each question is answerable by observing the image alone, without needing to reason about how something was “described” or “mentioned.”
>
>     In short, although textual artifacts may appear in internal steps of our pipeline, they are completely eliminated from the final evaluation format. We will clarify this in the revision and include an example in the appendix illustrating this transformation.

---

> ### Author Response · Authors · 2025-08-05
> **Kind Reminder: Your Additional Thoughts on Paper #9063 Are Welcome**
>
> Dear Reviewer  ysLX,
>
> Your expert feedback is crucial to refining this work. While we fully understand the discussion period may pose challenges for your schedule, we would value the chance to clarify any final points with you prior to its conclusion on Aug 8.
>
> We hope we've been able to address your questions and concerns so far. We would be glad to address any further concerns you may have, and we will try our best to clarify promptly.
>
> Thank you again for your feedback and comments; they were really helpful!
>
> Warm Regards, Authors of Submission #9063

---

> > ### Comment · Reviewer_ysLX · 2025-08-08
> >
> > Thank you to the authors for their detailed analysis of my concerns. I believe making sure clarification of the performance gap across domains and ablation on design choices will strengthen the presentation of the final paper. Expanded analysis to Reviewer K4PJ is also appreciated.

---

> > > ### Author Response · Authors · 2025-08-08
> > > **Thanks for your reply!**
> > >
> > > Thank you for your time and valuable feedback! We sincerely appreciate your effort in reviewing our work and are grateful for your positive assessment. Your insights have helped strengthen the paper, and we will incorporate them to further refine the final manuscript.

---

### Official Review · Reviewer_driM · 2025-07-03

**Clarity:** 3
**Significance:** 2
**Originality:** 2
**Rating:** 4
**Confidence:** 4

**Summary:**

This paper works on fine-grained visual reasoning in Multimodal Large Language Models (MLLMs), particularly their tendency to hallucinate or miss small but meaningful visual differences (e.g., object presence, count, spatial changes). The authors identify two key limitations: insufficient training data and weak alignment objectives.

To address these, they propose:
1. Micro Edit Dataset (MED): A large-scale dataset (~50K image-text pairs) of minimally edited image pairs and aligned captions spanning 11 fine-grained semantic edit types.
2. Controlled Visual Editing Pipeline: Generates image pairs using editing models like Gemini Flash 2.0, and captions via Qwen-VL and Qwen3-32B.
3. Micro Edit Detection (MED) Benchmark: A 165-question benchmark across the same 11 edit types, designed to evaluate sensitivity to small differences.
4. Feature Consistency Regularization: A fine-tuning objective that encourages stable visual embeddings for minor edits, thereby improving semantic alignment.

They fine-tune open-source models (e.g., Qwen2-VL-7B, LLaVA) and show significant improvements on the MED benchmark and generalization to other tasks (e.g., POPE, MMStar, BLINK). Their best model (Qwen2.5-VL-7B) matches or surpasses GPT-4o and Claude 3.7 Sonnet on fine-grained reasoning tasks.

**Questions:**

See weaknesses.

**Ethical Concerns:**

["NO or VERY MINOR ethics concerns only"]

**Final Justification:**

Authors addressed some of my concerns in the rebuttal.

**Limitations:**

Yes

**Quality:**

3

**Strengths And Weaknesses:**

Strengths:
1. Well-motivated and timely problem: Hallucination in MLLMs is a known but under-addressed issue, and the paper identifies a specific failure mode and addresses it directly.
2. High-quality dataset creation: the MED dataset is carefully constructed using various tools for editing, filtering, and alignment. The dataset and benchmark can be a solid contribution.

Weaknesses:
1. Finetuning with specific learning objective on a specific dataset are expected to gain improvements, but it is not clear how the fine-tuned models perform on general benchmarks.
2. the MED-Real set only has 35 image pairs, which might to small to verify real-world applicability.
3. Other families of open-source models to be evaluated, e.g., BLIP-2 / BLIP-3.

---

> ### Author Rebuttal · Authors · 2025-07-30
>
> **Overall Response**
>
> Thanks for the insightful feedback! Based on these comments, we have added a new experiment on BLIP-3o, expanded the MED-Real set to 50 samples, and included 4 general benchmarks (POPE, MMStar, BLINK, MMVP) to demonstrate the generalizability and robustness of our method. We also clarified the concerns regarding overfitting, dataset size, and model diversity. We will include all these discussions and results in the final version if the paper is accepted.
>
>
> > **Response to W1: Generalizability Beyond MED**
> >
>
> We thank the reviewer for raising the concern about generalization. As shown in Table 2 (Page 8) and the table below, our fine-tuned models consistently improve across diverse general benchmarks such as POPE, MMStar, BLINK, and MMVP.
>
> For instance, Qwen2.5-VL-7B improves in average score from 55.47 to 58.52, and MME from 1685.14 to 1701.87, demonstrating clear gains beyond MED. These results indicate that our fine-grained tuning approach enhances not only task-specific performance but also general multimodal reasoning, including coarse/fine perception, hallucination mitigation, and counting.
>
> | Model | Pope | Coarse | Fine | Visual_Sim | Visual_Corr | Count | MMVP | Ave | MME |
> | --- | --- | --- | --- | --- | --- | --- | --- | --- | --- |
> | Qwen2-VL-7B | 92.50 | 71.21 | 48.24 | 51.11 | 30.23 | 55.83 | 31.33 | 54.35 | 1679.52 |
> | Qwen2-VL-7B (Fine-tuned) | 96.27 | 73.92 | 46.16 | 51.85 | 33.72 | 59.17 | 32.67 | 56.25 | 1681.27 |
> | Qwen2.5-VL-7B | 96.29 | 73.95 | 57.35 | 49.63 | 33.72 | 50.00 | 27.33 | 55.47 | 1685.14 |
> | Qwen2.5-VL-7B (Fine-tuned) | 97.52 | 75.97 | 59.36 | 51.85 | 37.79 | 59.17 | 28.00 | 58.52 | 1701.87 |
> | LLaVA-V1.6-7B | 95.56 | 58.28 | 31.93 | 51.11 | 21.51 | 45.83 | 28.67 | 47.56 | 1441.89 |
> | LLaVA-V1.6-7B (Fine-tuned) | 97.39 | 56.74 | 35.13 | 48.14 | 24.42 | 49.17 | 30.00 | 48.71 | 1420.57 |
> | LLaMA-3.2-11B | – | 69.03 | 48.94 | 43.70 | 20.93 | 44.17 | 26.00 | 42.13 | 1421.71 |
> | LLaMA-3.2-11B (Fine-tuned) | – | 72.60 | 47.21 | 45.93 | 19.19 | 50.00 | 28.00 | 43.82 | 1430.67 |
>
> > **Response to W2: MED-Real Set Size and Generalization Validation**
> >
>
> We thank the reviewer for pointing out the limited size of the original MED-Real set (35 image pairs). While this subset already shows meaningful performance differences across models, we agree that further validation would be very helpful.
>
> To address this, we augmented the real-world evaluation with an additional 15 minimally edited image pairs, bringing the total to 50 samples. This allows us to test real-world generalization at multiple scales (15, 35, 50). The results are as follows:
>
> | Model | acc@15 | acc@35 | acc@50 |
> | --- | --- | --- | --- |
> | Qwen2 | 66.67 | 82.86 | 78.00 |
> | Qwen2 (Ours) | 73.33 | 82.86 | 80.00 |
> | Qwen2.5-3B | 46.67 | 54.29 | 52.00 |
> | Qwen2.5-3B (Ours) | 53.33 | 57.14 | 56.00 |
> | Qwen2.5-7B | 60.00 | 68.57 | 66.00 |
> | Qwen2.5-7B (Ours) | 73.33 | 74.29 | 74.00 |
> | LLaVA | 46.67 | 34.29 | 38.00 |
> | LLaVA (Ours) | 60.00 | 45.71 | 50.00 |
> | LLaMA3.2 | 26.67 | 40.00 | 36.00 |
> | LLaMA3.2 (Ours) | 46.67 | 51.43 | 50.00 |
>
> These results indicate:
>
> - Consistent performance boost from our fine-tuning across all sizes and models.
> - No overfitting to synthetic edits — the models trained on MED also generalize well to real-world minimal differences as Response to W1 explained.
>
> We will include the expanded 50-sample MED-Real set and analysis in the revised appendix to strengthen our empirical claims.
>
> > **Response to W3: Inclusion of Other Open-Source Models (e.g., BLIP-2 / BLIP-3)**
> >
>
> We appreciate the reviewer’s suggestion to evaluate additional model families beyond Qwen, LLaVA, and LLaMA.
>
> To address this, we include the newest model from BLIP series, **BLIP-3o** in our MED benchmark evaluation. The table below shows its performance across all 11 semantic edit types, both before and after fine-tuning on our dataset:
>
> | Model | Avg | Obj | Attr | Scene | Spatial | Action | Part | Count | Differ | Compar | Neg | Univ |
> | --- | --- | --- | --- | --- | --- | --- | --- | --- | --- | --- | --- | --- |
> | BLIP-3o-8B | 44.24 | 46.15 | 50.00 | 69.23 | 50.00 | 37.50 | 31.25 | 44.44 | 41.67 | 31.58 | 50.00 | 42.86 |
> | BLIP-3o-8B (Ours) | **50.90** | 38.46 | **62.50** | 46.15 | 42.86 | **62.50** | **37.50** | **66.67** | **50.00** | **52.63** | 35.71 | **57.14** |
>
> These results show:
>
> - Consistent performance gains from our fine-tuning approach.
> - Improved reasoning on complex types such as comparison, counting, and action.
>
> We will add these results to the revised paper and appendix to further demonstrate the broad applicability of our method across model families.

---

### Official Review · Reviewer_WxCP · 2025-07-03

**Clarity:** 4
**Significance:** 4
**Originality:** 3
**Rating:** 4
**Confidence:** 4

**Summary:**

This paper is targeting a core-issue in Large Vision-Language Models(LVLM) -- Fine-grained understanding of image content. First, the authors have shown that even in some most advanced LVLMs, such as GPT4o, there still exists issues with detail understanding. And they conclude these errors into a range of types. Then the authors construct a dataset for fine-grained image learning by their proposed pipeline. This pipeline uses image-editing to subtly modify two images and ask questions related to the original and edited images. Then the author argued that the models trained to describe the differences between two similar images are not doing well in the downstream description task. Inspired by this observation, the author proposed a new supervision training method to overcome this problem.  In this way, the model can learn to discreminate the fine-grained differences. The author also proposed a new loss function wich included the contrastive loss to train the LVLMs and show the performance superiority over other methods.

**Questions:**

1. In the data generation process, the pipeline uses a QwenVL model to generate instruction and filtering the images from the raw dataet. Then, the authors use the gemini model to generate the image pairs from the generated instructions. Since the instruction generation process is very important for the quality of generated images in the proposed MED dataset, do you consider any other quality control method for the instructions. From my understanding, QwenVL is indeed very powerful, but is far from perfect. A Qwen2.5VL-72B model can also have hallucination during generating instructions.

2. The authors use another data filtering module powered by CLIP models to varify the generated images quality compared to the original images. However, from my experiences with many image generative models, only semantic filtering is not enough for generated image quality control. In fact, the author should consider introducing some basic image quality evaluation method to control the qulity(i.e. SSIM). Furthermore, the authors only do CLIP embedding similarity check between the original image and the edited image but not between the edited image and the instruction. Why is that?

**Ethical Concerns:**

["NO or VERY MINOR ethics concerns only"]

**Final Justification:**

I am satisfied with the response and recommend to accept this paper. Though I have considered about rasing my rating, I don't feel this paper is so good for a 5, accept. So, I will maintain my score as 4.

**Limitations:**

Yes

**Quality:**

3

**Strengths And Weaknesses:**

Weaknesses:
1. This data generation pipeline involved too many models: from open source to close source, from LVLM to LLM; acrossing different platform; including text generation and image generation models. This whole pipeline is quite complicate and thus hard to replicate your work because of the involving of too many models.
2. Though I personally do not have any problem with leveraging synthetic data in training models, I still doubt the quality of the proposed datasets. I think the authors should release more quantitative results to evluate their prompt template and instruction quality.


Strength:
1. For the MED-Benchmarks, this work manually verified and categorize the question types. They also do the examination of data contamination for the splits.
2. This work does not simply train a downstream models on the curated datasets but verified the generalizing problem and minimize the performance gap in downstream tasks by proposing a new training method. The authors suggest that the performance gap of the existence LVLMs might be caused by the noisy training data which has a binomial probability distribution for giving a good quality description of the paired images.  And thus this work proposing a clean supervision training pipeline to mitigate such noise.
3. The curated dataset is quite valuable to the research community.

---

> ### Author Rebuttal · Authors · 2025-07-30
>
> **Overall Response**
>
> Thanks for the insightful feedback! Based on these comments, we clarified the modular design of our pipeline, addressed concerns about hallucination and instruction quality, and added quantitative metrics (e.g., MATTR, SSIM) to support our claims. We also highlighted the roles of Qwen3 and manual verification in ensuring data quality. All clarifications will be incorporated into the revised version.
>
>
> > **Response to W1: Complexity of the Data Generation Pipeline**
> >
>
> We thank the reviewer for raising this important concern regarding the complexity and reproducibility of our data generation pipeline. While our framework may appear to involve a large number of models at first glance, we would like to clarify that the core pipeline relies primarily on just two key components:
>
> 1. **Qwen2.5-VL-72B** — used consistently for instruction generation, caption refinement, and caption alignment.
> 2. **Gemini Flash 2.0** — used for high-fidelity image editing with minimal semantic changes.
>
> To ensure quality and robustness, we decompose the generation process into several *small, modular steps*, each focusing on a simple task well within the capability of the model employed. This modular design is intentional: rather than requiring complex, open-ended generation from a single model (which risks hallucination), each model performs only tightly-scoped operations, such as:
>
> - Revising captions only when elements are *visibly present* in the image.
> - Generating editing instructions based on predefined semantic categories (e.g., spatial, object, attribute).
> - Producing updated captions *conditioned on known differences*.
>
> This design minimizes the cognitive load on the model and enhances output reliability.
>
> Additionally, our dataset construction relies on publicly available APIs or open-source models, where all the scripts to leverage those toolkits are also released for reproducibility:
>
> - All prompt templates and model configurations have been released (in supplementary material).
> - Our dataset construction relies on publicly available APIs or open-source models, where all the scripts are provided.
> - Each pipeline stage is deterministic and documented in detail (see Section 3.2).
>
> We believe this strategy offers a principled balance between *model efficiency*, *data quality*, and *reproducibility*. We will clarify this point further in the revision to dispel the impression of an overly entangled system.
>
> > **Response to Q1: Instruction Quality and Hallucination Control in Qwen2.5-VL**
> >
>
> We appreciate the reviewer’s concern regarding potential hallucinations in instruction generation using Qwen2.5-VL-72B. While no vision-language model is entirely immune to hallucination, our pipeline is explicitly designed to *mitigate and control such risks* through two key strategies: task decomposition and multi-stage filtering.
>
> **1. Task decomposition: keeping model outputs focused and grounded**: Rather than asking the model to generate complex captions or instructions in one pass, we break the process into atomic and visually grounded subtasks, each scoped narrowly enough to reduce hallucination likelihood:
>
> - In Step 1, Qwen2.5-VL is only asked to *revise or extend* an existing caption, *conditioned on a clearly visible target object or attribute* and the original image. This minimizes hallucination by anchoring generation on visual evidence.
> - In Step 2, when generating the edited caption, the model receives the full original caption, the edit type (e.g., “change in spatial relation”), and the edited image. This instruction framing ensures that only relevant semantic changes are included.
> - In Step 3, difference descriptions are generated by a *separate text-only LLM* (Qwen3-32B), which compares the original and edited captions in isolation from visual noise, further reducing compounding hallucinations.
>
> **2. Multi-stage filtering: catching imperfect instructions or captions**: Beyond carefully-scoped prompts, we apply multiple steps of filtering to ensure quality and remove any hallucinated or inconsistent outputs:
>
> - In the initial image filtering stage, Qwen2.5-VL-72B evaluates each image-caption pair using a *structured scoring rubric* that checks for caption clarity, visual relevance, and editability. Only high-quality samples are retained.
> - After image editing and caption generation, we apply CLIP-based similarity filtering between the original and edited images to ensure that visual changes are minimal and semantically plausible.
> - For the final dataset, we manually verify 1,000 randomly sampled image pairs (spanning all 11 edit types). This human check helps catch rare but systematic hallucinations and refine our instruction prompt templates accordingly.
>
> Through this design, we ensure that Qwen2.5-VL is always solving a *constrained*, visually grounded task rather than performing free-form generation. We will clarify this design in the revised version and include additional metrics (e.g., instruction acceptance rates, manual error rates) to support our filtering effectiveness.
>
> > **Response to W2: Evaluating Prompt Template and Instruction Quality**
> >
>
> We appreciate the reviewer’s concern regarding the quality of the proposed synthetic dataset, especially the prompt templates and instructions.
>
> 1. **Quantitative metrics for text diversity:** To further validate textual quality, we calculated standard diversity metrics on our instructions:
>
>     - Moving Average TTR (MATTR, window size=50, ours):  0.7393
>     - Moving Average TTR (MATTR, window size=50, llava-instruct-150k): 0.7834
>
>     These scores indicate rich lexical diversity and low redundancy in instruction and prompting phrasing for a VQA dataset.
>
> 2. **Instruction quality control via Qwen3 and manual verification:** In our original data generation pipeline (Section 3.2), we explicitly employed Qwen3-32B, a strong text-only LLM, to refine and optimize edit instructions and difference descriptions. These instructions were rewritten for clarity, specificity, and semantic correctness. Furthermore, we conducted manual verification on 1,000 samples spanning all 11 edit categories to ensure alignment between images, captions, and difference descriptions, reducing systematic errors.
>
> 3. **Strong downstream performance:** The consistent improvements achieved across multiple model families (Qwen, LLaVA, LLaMA) and benchmarks demonstrate the practical value of the curated data.
>
> We will include these quantitative metrics and clarify the role of Qwen3 and manual validation in the revised version to strengthen transparency around instruction quality.
>
> > **Response to Q2: On Image Quality Filtering and Use of CLIP vs. SSIM**
> >
>
> We appreciate the reviewer’s insightful suggestions regarding image quality control.
>
> Our current pipeline adopts CLIP-based embedding similarity—a practice inspired by [1]—to ensure that edited images remain semantically close to the originals. CLIP excels at capturing high-level conceptual consistency, which aligns well with our goal of fine-grained semantic editing.
>
> To complement this, we computed SSIM on our 50K image pairs to assess low-level visual similarity:
>
> **SSIM Statistics (on MED Dataset):**
> Mean: 0.5011 | Std: 0.2028 | Min: -0.0134 | Max: 0.9839
>
> These results confirm that most edits are structurally subtle, as intended.
>
> We chose CLIP over SSIM because:
>
> - SSIM is known to struggle with localized edits on uniform backgrounds [2].
> - CLIP is more robust to superficial visual changes and focuses on semantic-level alignment.
>
> As for instruction-image alignment, the instructions are already used as generation inputs to Gemini and further validated via manual checks on 1,000+ samples. While CLIP-text similarity could be added, our current design already enforces semantic alignment both procedurally and manually.
>
> We will add the SSIM results to the appendix and clarify this in the revision.
>
> [1] Tong S, Liu Z, Zhai Y, et al. Eyes wide shut? exploring the visual shortcomings of multimodal llms[C]//Proceedings of the IEEE/CVF Conference on Computer Vision and Pattern Recognition. 2024: 9568-9578.
>
> [2] Ghazouali S E, Michelucci U, Hillali Y E, et al. CSIM: A Copula-based similarity index sensitive to local changes for Image quality assessment[J]. arXiv preprint arXiv:2410.01411, 2024

---

> > ### Comment · Reviewer_WxCP · 2025-08-04
> >
> > Thank you for the response and hardwork. While most of my concerns are addressed, I have an add-up question regarding to the CLIP similarity check. How do you decide the threshold of the CLIP-score for similarity check?

---

> > > ### Author Response · Authors · 2025-08-05
> > > **Kind Reminder: Your Additional Thoughts on Paper #9063 Are Welcome**
> > >
> > > Dear Reviewer WxCP,
> > >
> > > Thank you so much for your valuable comments and for taking time to review our rebuttal! While we fully understand the discussion period may pose challenges for your schedule, we would value the chance to clarify any final points with you prior to its conclusion on Aug 8.
> > >
> > > We hope we've been able to address your questions and concerns so far. We would be glad to address any further concerns you may have, and we will try our best to clarify promptly.
> > >
> > > Thank you again for your feedback and comments; they were really helpful!
> > >
> > > Warm Regards, Authors of Submission #9063

---

> ### Author Response · Authors · 2025-08-04
> **Response to Add-up Question : Threshold Selection in MED-Bench and MED Dataset**
>
> Thank you for your kind acknowledgment! We're glad our responses clarified your concerns.
>
> Regarding your follow-up question on CLIP-score thresholds, we clarify two distinct approaches:
>
> 1. For MED-Bench, we follow Tong et al. [1] in using a 95% similarity threshold.
>
> 2. For MED dataset construction, we conducted additional ablation studies during rebuttal to optimize the data quality-scale trade-off. Our methodology and key findings are summarized below:
>
> *Additional Ablation Study: Threshold (δ) vs. Performance*
>
> We trained identical Qwen2.5-VL-7B models on MED subsets filtered at different CLIP-similarity thresholds (δ), evaluating accuracy across 12 categories. Results are summarized below (also in W2 response for Reviewer WxCP):
>
> | Model with different δ | Data size | avg | obj | attr | scene | spatial | action | part | count | differ | compar | neg | univ |
> | --- | --- | --- | --- | --- | --- | --- | --- | --- | --- | --- | --- | --- | --- |
> | δ=0.7 | 50k | **51.51** | 57.14 | 56.25 | 53.84 | 42.86 | 46.67 | 43.75 | 55.56 | 50.00 | 47.37 | 50.00 | 64.29 |
> | δ=0.7 | random-12k | 41.21 | 53.85 | 43.75 | 53.85 | 42.86 | 62.50 | 18.75 | 22.22 | 33.33 | 31.58 | 50.00 | 50.00 |
> | δ=0.8 | 36k | 46.67 | 53.84 | 37.50 | 46.15 | 21.43 | 43.75 | 31.25 | 61.11 | 50.00 | 42.11 | 50.00 | 78.57 |
> | δ=0.8 | random-12k | 42.42 | 61.54 | 31.25 | 53.85 | 28.57 | 37.50 | 37.50 | 33.33 | 41.67 | 36.84 | 64.86 | 50.00 |
> | δ=0.9 | 12k | 44.85 | 46.15 | 50.00 | 46.15 | 35.71 | 43.75 | 43.75 | 38.89 | 50.00 | 31.58 | 57.14 | 57.14 |
>
> - Higher thresholds (δ≥0.8) improved per-sample alignment (e.g., δ=0.9 achieved 78.6% in universal tasks) but reduced data size (12k samples), causing significant drops in compositional tasks like spatial reasoning (-7.2% vs δ=0.7) and counting (-16.7%).
> - Lower thresholds (δ<0.7) introduced noisy pairs, degrading performance across all categories.
> - δ=0.7 (50k samples) delivered peak average accuracy (51.5%) by:
>     - Preserving critical diversity for complex reasoning (e.g., +18.8% in attribute binding vs δ=0.8)
>     - Maintaining scale advantages (random 12k subset at δ=0.7 underperformed by 10.3%)
>
> This threshold maximizes robustness while avoiding artificial scale limitations. We will detail this analysis in Section 4.2 (revised manuscript) and appreciate your insightful query. Welcome any additional questions should you wish to discuss this further!
>
> [1] Tong S, Liu Z, Zhai Y, et al. Eyes wide shut? exploring the visual shortcomings of multimodal llms[C]//Proceedings of the IEEE/CVF Conference on Computer Vision and Pattern Recognition. 2024: 9568-9578.

---

> > ### Comment · Reviewer_WxCP · 2025-08-05
> >
> > Thank you for your resposne. I have no follow up questions and would consider raise my rating.

---

> > > ### Author Response · Authors · 2025-08-05
> > > **Thanks for your reply!**
> > >
> > > Thank you so much for your time and valuable feedback. We truly appreciate your consideration in raising the rating! Your suggestions will be carefully incorporated to further improve our manuscript in the final version. We'd be happy to discuss anything further :)

---

### Note · Authors · 2025-08-14

We sincerely thank the reviewers for their thoughtful feedback and constructive suggestions.

### **Key strengths highlighted by reviewers**

- **Well-motivated problem:** *[Reviewer WxCP, Reviewer driM, Reviewer ysLX, Reviewer PV47]* recognized our focus on hallucination from fine-grained visual differences as timely and underexplored.
- **High-quality dataset and benchmark:** *[Reviewer WxCP, Reviewer driM, Reviewer PV47, Reviewer K4PJ]* appreciated the design of MED and MED-Bench, with semantic diversity, manual checks, and systematic filtering.
- **Principled method with strong results:** *[Reviewer WxCP, Reviewer driM, Reviewer ysLX, Reviewer K4PJ]* highlighted the effectiveness of our feature consistency loss and fine-tuning across fine-grained and general VL tasks.
- **Broad applicability:** *[Reviewer driM, Reviewer PV47]* noted generalization across model families (Qwen, LLaVA, LLaMA) and improved real-world performance.

### **Addressing reviewer concerns**

We have revised the manuscript to address the concerns:

- **Pipeline complexity and reproducibility:** *[Reviewer WxCP, Reviewer PV47]* We clarified the roles of Qwen2.5-VL, Gemini Flash 2, and CLIP in our modular pipeline, and released full code, prompts, and configs.
- **Instruction quality, filtering, and annotation:** *[Reviewer WxCP, Reviewer ysLX]* We reported MATTR diversity, 93.8% verified instruction accuracy, and detailed Qwen3-32B’s role in reducing hallucinations. SSIM analysis and CLIP threshold ablations (δ = 0.7) were added.
- **Benchmark design and generalization:** *[Reviewer ysLX, Reviewer PV47]* We explained the caption-to-QA transformation, verified no text artifacts, and analyzed pretraining-driven synthetic–real gaps, showing strong transfer for open-source models.
- **Real-world robustness:** *[Reviewer driM, Reviewer ysLX, Reviewer K4PJ]* We expanded MED-Real (35→50) and presented results on POPE, MMStar, BLINK, and MMVP, showing consistent gains over baselines.
- **Model diversity:** *[Reviewer driM, Reviewer K4PJ]* We added BLIP-3o results, confirming generality beyond Qwen and LLaVA.
- **Ablations and scaling:** *[Reviewer ysLX, Reviewer PV47]* We studied CLIP thresholds, edit diversity, and backbone scaling, validating dataset and method choices.

Reviewers confirmed most concerns were resolved, with some concerning a score increase. We believe the revisions strengthen both the scientific contribution and the clarity of our work.

---

### Decision · Program_Chairs · 2025-09-17

**Decision:**

Accept (poster)

**Comment:**

Summary: Introduces MED (50K minimally edited pairs) + MED-Bench, and an SFT feature-consistency objective to improve fine-grained difference detection and reduce hallucinations.  Also gives gains on POPE/MMStar/BLINK/MME vs strong baselines.

Strengths: timely focus on fine-grained hallucination, carefully filtered dataset/benchmark with manual checks, and empirical gains with released code/prompts/configs.

Weaknesses: complex multi-model pipeline with concurrent reproducibility burden.  Small real-image set and initially narrow model coverage.

(d) clear dataset+benchmark contribution and an effective lightweight training signal that measurably reduces hallucination across tasks/models.

(e) Discussion/rebuttal summary: reviewers pressed on pipeline complexity, instruction/annotation quality, model diversity, and caption artifacts; authors responded to address many concerns.